**Primary and Secondary Ice Production: Interactions and Their**
**Relative Importance**
Xi Zhao[1] and Xiaohong Liu[1]
[1]Department of Atmospheric Sciences, Texas A&M University, College Station, Texas 77840, USA
*Correspondence to*: Xiaohong Liu (xiaohong.liu@tamu.edu)
**Abstract**
A discrepancy of up to 5 orders of magnitude between ice crystal and ice nucleating
particle (INP) number concentrations was found in the measurements, indicating the
potential important role of secondary ice production (SIP) in the clouds. However, the
interactions between primary and SIP processes and their relative importance remain
unexplored. In this study, we implement five different ice nucleation schemes as well as
physical representations of SIP processes (i.e., droplet shattering during rain freezing,
ice-ice collisional break-up, and rime splintering) in the Community Earth System Model
version 2 (CESM2). We run CESM2 in the single column mode for model comparisons
with the DOE Atmospheric Radiation Measurement (ARM) Mixed-Phase Arctic Cloud
Experiment (M-PACE) observations.
We find that the model experiments with aerosol-aware ice nucleation schemes and
SIP processes yield the best simulation results for the M-PACE single-layer mixed-phase
clouds. We further investigate the relative importance of ice nucleation and SIP to ice
number and cloud phase as well as interactions between ice nucleation and SIP in the M-
PACE single-layer mixed-phase clouds. Our results show that SIP contributes 80% to the
total ice formation and transforms ~30% of pure liquid-phase clouds simulated in the
model experiments without considering SIP into mixed-phase clouds. SIP is not only a
result of ice crystals produced from ice nucleation, but also competes with the ice
nucleation by reducing the number concentrations of cloud droplets and cloud-borne dust
INPs. Conversely, strong ice nucleation also suppresses SIP by glaciating mixed-phase
clouds and thereby reducing the amount of precipitation particles (rain and graupel).

# 1 Introduction

Ice crystals significantly impact microphysical and radiative properties of mixed-phase clouds (Korolev and Isaac 2003; Korolev et al., 2017; Morrison et al., 2012), which further impact the Earth's energy budgets. Ice particles in mixed-phase clouds with temperatures between about -38 ℃ and 0 ℃ can be formed via heterogeneous ice nucleation on ice nucleating particles (INPs) or arisen through secondary ice production (SIP) (Kanji et al., 2017; Field et al., 2017). Ice crystals that fall from overlying cirrus clouds can provide another source of ice in mixed-phase clouds. There are three identified heterogeneous ice nucleation mechanisms, namely, contact, deposition, and immersion/condensation freezing. Dust is generally considered as the most effective INPs for heterogeneous ice nucleation at temperatures below about -15 ℃ (Hoose et al., 2008; Atkinson et al., 2013; Kanji et al., 2017). SIP processes generate additional ice crystals, often involving the primary ice. Several SIP mechanisms have been suggested: rime splintering (also known as the Hallett–Mossop (HM) process), droplet shattering during rain freezing (FR), ice-ice collisional break-up (IIC), and fragmentation during the sublimation of ice bridge (Field et al., 2017; Korolev et al., 2020). In addition, other microphysical processes such as rain formation, ice growth, and ice sedimentation are important for mixed-phase cloud properties (Mülmenstädt et al., 2021; Tan and Storelvmo, 2016). Regarding ice-related microphysical processes in mixed-phase clouds, some processes, including riming, accretion, and the Wegener–Bergeron–Findeisen

(WBF) process can increase the ice mass mixing ratios while have no effect on ice crystal
number concentrations (ICNCs). On the other hand, some processes such as ice
aggregational growth decrease the ICNCs while have no impacts on the ice mass mixing
ratios.
A systematically measured discrepancy by up to 5 orders of magnitude between the
ICNCs and INP number concentrations has been reported in previous studies (Mossop,
1985; Lasher-Trapp et al., 2016; Field et al., 2017), indicating the existence of additional
ice production mechanisms in addition to the primary ice production (PIP) or ice
nucleation. Moreover, a strong increase in ICNCs over INP number concentrations may
suggest that the PIP would be less important once the SIP processes take place in the
clouds. However, the relative importance between PIP and SIP to the ice formation in
mixed-phase clouds is largely unknown and warrants a further investigation.
Previous studies have identified the potential role of PIP in initiating the SIP based
on measurements and idealized parcel model simulations. Sullivan et al. (2018) found
that clouds with INP concentrations from 0.002 to 0.15 $L^{-1}$ can initiate the IIC
fragmentation to produce enough ice crystals based on parcel model simulations. They
also indicated that higher INP concentrations enhance the IIC and HM process rates,
while the FR rate is not dependent on the INP concentration. Huang et al. (2017)
suggested that a number concentration as low as 0.01 $L^{-1}$ for primary ice is sufficient to
generate secondary ice though the HM process in the cumulus clouds observed over the
British Isles during the Ice and Precipitation Initiation in Cumulus (ICEPIC) campaign.
Crawford et al. (2012) found that a small amount of primary ice (0.01 $L^{-1}$) could produce
enough ice crystals with concentrations up to 100 $L^{-1}$ through the SIP processes in a
shallow convective cloud over the UK. Beard (1992) found that the droplet shattering can
be initiated by primary ice with a number concentration of ~0.001 $L^{-1}$ in the
measurement of a warm-base convective cloud. Despite the above progress, many
questions remain unexplored for the Arctic mixed-phase stratus clouds, e.g., whether PIP
always promotes the SIP and how SIP influences the PIP.
SIP is not only a result of PIP, but also can interact with and may even suppress the
subsequent PIP. A previous study indicated a 40% decrease of heterogeneous ice
nucleation after implementing the SIP into a model (Phillips et al., 2017b), because some
of the mixed-phase clouds with weak ascents and low humidities are fully glaciated and
become ice-only phase. The influence of SIP processes on PIP is far less investigated
compared to the limited studies of PIP influence on the SIP.
The goal of this study is to investigate the relative importance of PIP and SIP to
ICNCs and their interactions in the Arctic mixed-phase stratus clouds. We are attempting
to address the following scientific questions: Is the PIP still important for ICNCs once the
SIP processes take place? What effect does the PIP have on the SIP processes? Once
happening, how do the SIP processes affect the following PIP through the cloud
microphysical processes? This paper is organized as follows. Section 2 introduces the
model and the model parameterizations we used in this study. Section 3 describes the
model setup and model experiments. Section 4 presents the model results and comparison
with observations. The main findings of this study are summarized in section 5.

## 94    2 Model and Parameterizations

### 95    2.1 Model description

This study uses the Community Atmosphere Model version 6 (CAM6), the

atmosphere component of the Community Earth System Model version 2 (CESM2)
(Danabasoglu et al., 2020) for all the model experiments. In CAM6, the cloud
microphysics is represented by the version 2 of a double-moment scheme (Gettelman and
Morrison, 2015, hereafter as MG2), which predicts mass mixing ratios and number
concentrations of four categories of hydrometeors: cloud droplet, cloud ice, rain, and
snow. Graupel is not considered in the default CAM6 with MG2 microphysics.
Furthermore, the MG scheme only treats the HM process among various SIPs. The
aerosol properties and processes are represented by the four-mode version of the Model
Aerosol Module (MAM4) (Liu et al., 2012, 2016). Ice nucleation in cirrus clouds
considers the homogeneous freezing of sulfate droplets and heterogeneous freezing on
dust (Liu and Penner, 2005), while the classical nucleation theory (CNT) is used to treat
the heterogeneous ice nucleation in mixed-phase cloud regime (Wang et al., 2014; Hoose
et al., 2010).
In our previous study (Zhao et al., 2021a), we have implemented the
parameterizations (Phillips et al., 2017a, 2018) of the two new SIP processes: FR and IIC
(without graupel involved) into CAM6 via an emulated bin framework. The graupel
related IIC was further included in CAM6 (Zhao and Liu, 2021), with the graupel amount
diagnosed following Zhao et al. (2017). In this study, we compare several different ice
nucleation schemes in CAM6 to examine the relative importance and interactions
between PIP and SIP in the Arctic mixed-phase clouds.

## 118 2.2 Ice nucleation parameterization

**CNT scheme**
The default CAM6 uses the CNT for treating the ice nucleation in mixed-phase
clouds. CNT is a "stochastic" scheme which calculates the ice nucleation rates from
deposition, contact, and immersion freezing of cloud droplets, depending on the surface
areas and contact angles of cloud-borne dust and black carbon (BC) particles. The contact
angle is used as a proxy for the ice nucleation efficiency on INPs. CNT is formulated
based on Hoose et al. (2010) and implemented in CAM by Wang et al. (2014) with
further improvements of using a probability density functions (PDF) of contact angle
instead of a single contact angle in Hoose et al. (2010).

**N12 scheme**
Based on laboratory measurements from the Aerosol Interaction and Dynamics in
the Atmosphere (AIDA) cloud chamber, Niemand et al. (2012) (hereafter as N12)
proposed a surface-active site density-based scheme for the immersion freezing of cloud
droplets on dust aerosols. N12 is an empirical scheme that connects the dust INP number
concentration to the density of ice-active surface sites ($n_s(T)$) at a given temperature $T$
(K), total number concentration of dust aerosols ($N_{tot}$, L$^{-1}$), and dust particle surface area
($S_{ae}$, m$^2$). The dust INP number concentration (L$^{-1}$) in N12 is calculated as:
$$N_{INP}(T) = N_{tot}S_{ae}n_s(T) \qquad (1)$$
in which $S_{ae}$ is calculated based on the dry diameter of dust particles, and $n_s(T)$ (m$^{-2}$) is
calculated following:
$$n_s(T) = e^{(-0.517(T-273.15)+8.934)} \qquad (2)$$

**D15 scheme**
An empirical scheme for the immersion freezing of cloud droplets on dust aerosols
was developed by considering dust particles with sizes larger than 0.5 μm (DeMott et al.,
2015), hereafter referred to as D15. This scheme argues that dust particles smaller than
0.5 μm may not be efficient INPs (DeMott et al., 2010, 2015). D15 was developed as a
combination of field campaign and laboratory data measured by the continuous flow
diffusion chamber (CFDC) and the Aerosol Interactions and Dynamics of the
Atmosphere (AIDA) cloud chamber. The field campaign data were obtained during the
2007 Pacific Dust Experiment (PACDEX) on the NSF/NCAR G-V aircraft over the
Pacific Ocean basin (Stith et al., 2009), and the 2011 Ice in Clouds Experiment – Tropical
(ICE-T) on the NSF/NCAR C-130 aircraft flown from St. Croix, US Virgin Islands
(Heymsfield and Willis, 2014). The dust INP number concentration (std L$^{-1}$) in D15 is
calculated as:
$$N_{INP}(T) = a(n_{0.5})^b e^{c(T-273.15)-d} \tag{3}$$

in which $n_{0.5}$ is the number concentration (std cm$^{-3}$) of dust particles with diameters
larger than 0.5 μm, and the parameters $a = 3$, $b = 1.25$, $c = -0.46$, and $d = 11.6$.

**B53 scheme**

Bigg (1953) proposed a volume-dependent immersion freezing scheme, hereafter

referred to as the B53 scheme. In this scheme, the number concentration of frozen cloud
droplets with a diameter $D$ is given as:
$$\frac{\partial N_{B53}}{\partial t} = N_c(D) \times \left(-B \times \left(e^{A \times (T_0 - T)} - 1\right) \times \frac{\pi D^3}{6}\right) \tag{4}$$

in which $\frac{\partial N_{B53}}{\partial t}$ is the ice number production rate ($kg^{-1}s^{-1}$), $T$ is the environmental
temperature in unit of K, $T_0 = 273.15$ K, $A = 0.66$ and $B = 100$, and $N_c(D)$ is the number
mixing ratio of cloud droplets ($kg\ kg^{-1}$) with a diameter $D$.

**M92 scheme**
An empirical temperature dependent scheme was developed based on measurements
in the Northern Hemisphere midlatitudes by using a continuous-flow diffusion chamber
(CFDC) (Meyers et al., 1992), hereafter referred to as M92. The INP number
concentration ($L^{-1}$) is calculated as:
$$N_{INP} = e^{a+b\times\left(\frac{e_{sl}-e_{si}}{e_{si}}\right)} \qquad (5)$$
in which $a = -0.639$, $b = 0.1296$, and $e_{sl}$ and $e_{si}$ are the saturation vapor pressures with
respect to liquid and ice, respectively.
Marine organic aerosols and sea salt are not included as INPs in any of the above ice
nucleation parameterizations.

## 179    2.3 Graupel parameterization

The graupel mass mixing ratio ($q_g$) is diagnosed as precipitation ice mass (currently
snow, $q_s$) multiplied by the rimed mass fraction $Ri$ (Zhao et al., 2017),
$$q_g = q_s \times Ri \qquad (6)$$
The rimed mass fraction $Ri$ is calculated as:
$$Ri = \frac{m_{rimed}}{m_{rimed}+m_{unrimed}} \approx \frac{1}{1+\frac{6\times10^{-5}}{q_c(q_i+q_s)^{0.17}}} \qquad (7)$$
q_c, q_i, and q_s in (7) are modeled cloud water, cloud ice, and snow mixing ratios
(kg kg$^{-1}$), respectively. The graupel number is assumed to have the same ratio to
snow number as the ratio of graupel mass to snow mass.

# 3 Model setup, experiments, and observations

The CAM6 model was set up with the Single Column Atmospheric Model (SCAM)
configuration. SCAM is an efficient approach to understand the physical processes in the
model without the impact from nonlinear interactions with dynamic processes (Gettelman
et al., 2019a). In SCAM, aerosols are initialized with monthly averaged profiles for
different aerosol types (sulfate, BC, particulate organic matter, secondary organic aerosol,
dust, sea salt) at a given location, which are derived from a present-day CAM6
climatological simulation. Aerosol processes are fully represented in SCAM, including
emission, transport, chemistry, dry and wet scavenging, and aerosol-radiation and
aerosol-cloud interactions (Liu et al., 2012; 2016). For example, the interstitial aerosols
will be activated to become the cloud-borne aerosols once cloud droplets are nucleated in
the cloud microphysics. The cloud-borne aerosols will be released to the interstitial
aerosols once cloud droplets evaporate, which can be re-activated when cloud droplets
are nucleated. The simulated aerosols are relaxed to a monthly averaged profile, and
temperature and horizontal winds to the large-scale forcing data every three hours. More
details about the model setup and the large-scale forcing data used to drive the model
experiments can be found in Zhao et al. (2021a).
This study focuses on the Arctic mixed-phase clouds observed during the
Department of Energy (DOE)'s Atmospheric Radiation Program (ARM) Mixed-Phase
Arctic Cloud Experiment (M-PACE), which was conducted in the North Slope of Alaska
in October 2004 (Verlinde et al., 2007). Four major cloud regimes were identified during
M-PACE, i.e., the multilayer stratiform cloud period (6 to 8 October 2004), the single-
layer boundary-layer stratiform cloud period (9 to 12 October), the transition cloud
period (16 October), and the frontal cloud period (18 to 20 October).

Several SCAM model experiments are conducted in this study (Table 1), covering

the whole M-PACE period from 5 to 22 October 2004. The CNT experiment uses the
default CAM6 model with the MG scheme, in which only HM is considered for SIP. The
ice nucleation is treated by the CNT scheme. The N12, D15, B53, and M92 experiments
are the same as the CNT experiment except using the respective ice nucleation scheme to
replace the CNT scheme for the immersion freezing (section 2.2). The deposition and
contact ice nucleation are still based on the CNT scheme in the N12 and D15
experiments, and based on Meyers et al. (1992) and Young (1974), respectively in the
B53 and M92 experiments. The impacts of other SIP mechanisms in addition to HM, i.e.,
FR and IIC, are addressed in the CNT_SIP experiment. To evaluate the SIP sensitivity to
ice nucleation, four additional experiments with different ice nucleation schemes are
conducted, and these experiments are named as N12_SIP, D15_SIP, B53_SIP, and
M92_SIP.

The model simulations are compared against the M-PACE observations. The ice

water path (IWP) and liquid water path (LWP) are based on ground-based remote sensing
observations provided by Zhao et al. (2012) with uncertainties within one order of
magnitude (Dong and Mace, 2003; Shupe et al., 2005; Deng and Mace, 2006; Turner et
al., 2007; Wang, 2007; Khanal and Wang, 2015). The INP concentrations are based on
in-situ observations by a CFDC on board an aircraft (Prenni et al., 2007). The ICNCs and
cloud phase are based on in-situ observations and provided by McFarquhar et al. (2007).
However, the ICNCs were measured before anti-shattering algorithms were developed to
remove the shattered particles for the 2DC cloud probe. To remove the shattering effect,
the M-PACE observed ICNCs were scaled by a factor of 1/4, as Jackson and McFarquhar
(2014) and Jackson et al. (2014) suggested an averaged reduction of ICNCs by 1–4.5
times in other field campaigns which adopted the anti-shattering algorithms and also used
the 2DC cloud probe. A different scaling factor of 1/2 is applied to the observed ICNCs,
which increases the observed ICNCs by a factor of 2 (Figure S3). The underestimation of
ICNCs by the model experiments with only ice nucleation (CNT, N12 and D15) is even
worse and our conclusion regarding model and observation comparison of ICNCs is not
changed. Since the measurements cannot distinguish snow from cloud ice, the simulated
ICNC, IWP, and IWC all include the snow component for the comparison with
observations.

# 4 Results

## 4.1 Overview of modeled clouds during M-PACE

The simulated LWP and IWP are compared with observations in Fig. 1 and Fig. S1.

First, SIP processes have a varied impact on modeled LWP and IWP, depending on ice

nucleation. In the SIP experiments with the CNT, N12, and D15 ice nucleation schemes,

simulated IWP is increased from 5 to 10 g m$^{-2}$ and LWP is decreased from 156 to 97 g m$^{-2}$

averaged over the M-PACE period after considering the SIP. In the SIP experiments with

the B53 and M92 schemes, however, SIP has a minimal impact on the LWP/IWP. Second,

the B53, B53_SIP, M92, and M92_SIP produce the largest IWP (~12 g m$^{-2}$ averaged over

the M-PACE period), followed by CNT_SIP, N12_SIP, and D15_SIP (~10 g m$^{-2}$ averaged

over the M-PACE period). CNT, N12, and D15 experiments produce the smallest IWP (~5

g m$^{-2}$ averaged over the M-PACE period). These characteristics are also evident in the

vertical profiles of LWC and IWC in Fig. 2 and Fig. S2. It indicates that the B53 and M92

nucleation schemes are highly efficient in forming ice; in comparison, the SIP simulations

using CNT/N12/D15 ice nucleation schemes show lower ice production capabilities. B53,

B53_SIP, M92, and M92_SIP experiments generate the closest IWP (~12 g m$^{-2}$ averaged

over the M-PACE period) compared with the observation (~64 g m$^{-2}$). However, these four

experiments also show substantially low biases of LWP (~40 g m$^{-2}$ compared with 126 g

m$^{-2}$ in the observation averaged over the M-PACE period). As shown in Fig. 1 and Fig. S1,

the mixed-phase clouds are almost fully glaciated during the single layer stratus period.

Therefore, the CNT_SIP, N12_SIP, and D15_SIP experiments give the best simulation
results in terms of LWP and IWP during the M-PACE. Adding the SIP does not change the
modeled LWP/LWC and IWP/IWC with the B53 and M92 ice nucleation schemes. On the
contrary, SIP decreases the LWP/LWC by 38% and doubles the IWP/IWC with the CNT,
N12, and D15 ice nucleation schemes.

## 4.2 PIP and SIP importance to ice number and cloud phase

A comparison between INP number concentrations ($N_{INPs}$) and ICNCs during 9-12
October is shown in Fig. 3. During this period, a long-lived single-layer mixed-phase cloud
occurred between 800-950 hPa, with observed cloud top temperatures of –17°C (Verlinde
et al., 2007). Modeled ICNCs include ice crystals of all sizes, since our purpose here is to
compare $N_{INPs}$ with ICNCs. With the empirical ice nucleation schemes (e.g., N12 and
D15), there appears an inversely relationship between $\log_{10}(N_{INPs})$ and temperature (Fig. 3c,
d). However, this relationship is not as clear with the CNT and B53 schemes, and $N_{INPs}$
reduces rapidly at temperatures warmer than -15 °C, from ~$10^{-1}$ L$^{-1}$ at –17°C to <$10^{-5}$ L$^{-1}$ at
–13°C (Fig. 3b, e). In contrast, $N_{INPs}$ with the aerosol-independent M92 scheme is less
variable with temperature, and is 1-7 orders of magnitude higher than that with the aerosol-
aware schemes, such as CNT, N12, and D15, particularly at warmer temperatures. We note
that the model may significantly underestimate dust burdens in the Arctic regions by 1-2
orders of magnitude (Shi and Liu, 2019) and may miss the representation of other INP
sources in the Arctic (e.g., local high-latitude dust, marine and terrestrial biological
aerosols).

The ice multiplication from the SIP processes can be noted by the results that modeled

ICNCs are higher than modeled $N_{INPs}$ in Fig. 3, even when we account for the 1-2 orders of
magnitude underestimation of $N_{INPs}$ for these aerosol-aware ice nucleation schemes (CNT,
N12 and D15). The model simulation with the aerosol-independent nucleation scheme M92
is an exception (Fig. 3f). However, M92, which was based on the measurements in the
Northern Hemisphere mid-latitudes may overestimate the $N_{INPs}$ in the Arctic during the M-
PACE (Prenni et al., 2007) (comparing $N_{INPs}$ in Fig. 3a, f). Observed $N_{INPs}$ are mostly
within the medium range of observed ICNCs (Fig. 3a). However, observed ICNCs only
include ice crystals with diameters larger than 100 μm, and thus the actual ambient ICNCs
including all-size ice crystals can be much higher.

Although these schemes differ in details about temperature and aerosol dependences

(Figure 3), CNT, N12, and D15 predict much lower INP concentrations during M-PACE
than those from the B53 and M92 schemes. With these low INP concentrations, the
single-layer clouds modeled with the CNT, N12 and D15 schemes have similar cloud
states (e.g., dominated by liquid-phase) (Figures 1 and 2). In contrast, B53 and M92
which are only dependent on temperature and not limited by aerosols predict much higher
INP concentrations. With these high INP concentrations, modeled clouds with the B53
and M92 schemes are dominated by ice-phase.

Figure 4 shows the vertical distribution of ICNCs in the single-layer mixed-phase

clouds during October 9 to 12 from model simulations and observations. Here, modeled
and observed ICNCs only include ice particles with diameters larger than 100 μm. The
observed ICNCs, which range mainly between 0.1 and 1 $L^{-1}$, show a slight decrease with
altitude. CNT, N12, and D15 all show rather constant ICNCs with altitude, which are also
one order of magnitude lower than the observation. The ICNCs with B53 and M92 are
increased compared with CNT, but the vertical ICNC patterns show increasing trends with
altitude. As suggested in Morrison et al. (2012), the long-lived Arctic mixed-phase clouds
are featured with liquid phase at cloud top and ice phase at cloud bottom. The SIP
experiments with CNT, N12, and D15 increase the ICNCs mainly in the lower portion of
clouds, and thus improve the agreement with the observed vertical distribution trend of
ICNCs. In contrast, SIP does little changes to the ICNCs when the B53 and M92 schemes
are used.
The ICNC in the CNT experiment and ice enhancement ratios of ICNC from the other
experiments to that from CNT are shown in Fig. 5. The enhancement ratios are around 1.0
in the N12 and D15 experiments, suggesting that these three ice nucleation schemes (CNT,
N12, and D15) produce similar magnitudes of ICNCs. Correspondingly, the ice
enhancement ratio patterns in the CNT_SIP, N12_SIP, and D15_SIP experiments show the
dominant role of SIP in increasing the ICNCs by up to 4 orders of magnitude. In contrast,
the ice enhancement ratios in B53 and M92 are up to 3.4 and 4 orders of magnitude,
respectively, suggesting that the B53 and M92 schemes are much more efficient in
producing ice particles than CNT, N12, and D15. The ice enhancements in B53_SIP and
M92_SIP are mainly contributed from the ice nucleation (B53 and M92) with only a minor
contribution from SIP, unlike the N12_SIP and D15_SIP experiments where the ice
enhancements are predominantly contributed from SIP.

Figure 6 shows the vertical distribution of the supercooled liquid fraction (SLF)

(defined as LWC/TWC, TWC = LWC + IWC) in the single-layer mixed-phase clouds
during October 9 to 12 from aircraft observations and model simulations. The CNT, N12,
and D15 experiments share the similar cloud phase distribution and all overestimate the
SLF in clouds with the vertically averaged SLF of 96.25%, 96.28%, and 96.26% in CNT,
N12, and D15, respectively, compared to 64.35% from the observation. On the contrary,
the B53 and M92 experiments with more efficient ice nucleation show predominantly ice
phase clouds with the vertically averaged SLF of 17.62% and 16.43%, respectively, which
agrees with previous findings (Liu et al., 2011). The experiments with SIP (CNT_SIP,
N12_SIP, and D15_SIP) improve the simulated cloud phase by reducing the SLF in the
CNT, N12, and D15 experiments, respectively, and the SLF patterns are also similar
among these experiments. SIP transforms ~30% of pure liquid-phase clouds simulated in
the CNT, N12, and D15 experiments into mixed-phase clouds. The TWC is reduced with
the total water path (TWP = LWP + IWP) decreased from 218.5, 219.2, and 219.1 in CNT,
N12, and D15 to 132.6, 131.0, and 130.8 in CNT_SIP, N12_SIP, and D15_SIP,
respectively. SIP does little changes to the cloud phase simulated in the B53_SIP and
M92_SIP experiments, since the clouds are already glaciated by ice crystals nucleated with
the B53 and M92 schemes. These findings highlight that the "foundation" effect of PIP on
the cloud phase. We note that the CNT_SIP, N12_SIP, and D15_SIP experiments overall
have the best performance in terms of vertical distribution of ICNCs and cloud phase
during the single-layer mixed-phase cloud period.

Figure 7 show the relative contributions from PIP and SIP processes to the total ice

mass production from model experiments with different ice nucleation schemes averaged
over different M-PACE periods. The ice mass production rates are calculated by
multiplying ice number production rates from parameterizations by the initial mass of an
ice particle ($2.093 \times 10^{-15}$ kg). We notice that the CNT_SIP, N12_SIP, and D15_SIP
experiments have similar relative contributions between PIP and SIP. The averaged PIP
contribution is around 20% for all the cloud types observed during M-PACE, with the
maximum contribution of 60% for the frontal clouds, and the minimum contribution of 7%
for the single-layer mixed-phase clouds. Moreover, the IIC is the dominant ice production
process in these three experiments, with an averaged contribution of 60%. On the contrary,
the B53_SIP and M92_SIP experiments show much larger contributions from PIP, which
contributes 65% and 80% to the total ice production, respectively averaged for all the cloud
types. However, we note that the unrealistic pure ice-phase clouds simulated in the B53
and M92 experiments imply that the role of ice nucleation in these experiments is
overstated. Given that the CNT_SIP, N12_SIP, and D15_SIP experiments give the best
performance in simulating ICNCs and cloud phase, their estimates of the relative
importance of primary and secondary ice production are more reliable.
Since the INP number concentrations in CNT, N12 and D15 are significantly lower
than the observations (Figure 3), a sensitivity test using the CNT scheme with increased
dust concentrations by 100 times shows overall similar cloud properties. However, the
relative contribution of primary ice nucleation to total ice production is increased by a
factor of ~2 to 30% averaged for all the cloud types and to 20% for the single-layer mixed-
phase clouds.

## 4.3 Interactions between PIP and SIP

Figure 8 shows the temporally-averaged vertical profiles of PIP and SIP process rates
for ice mass and total from experiments with the CNT and M92 ice nucleation schemes,
respectively during the single-layer mixed-phase cloud period (October 9 to 12). As shown
in Fig. 8a, clear suppression of PIP by SIP is revealed: the ice nucleation rate is reduced
after the SIP is introduced for both CNT and M92 ice nucleation but with different
sensitivities. The M92 ice nucleation is more suppressed by SIP than the CNT ice
nucleation. The peak PIP rate is reduced by about one order of magnitude in M92
compared to a factor of 3 in CNT. The suppression of PIP by SIP is robust for the other
three ice nucleation schemes over the single-layer mixed-phase cloud period (Fig. S5), as
well as for the whole M-PACE period (Figs. S6 and S7).

The mechanism for the suppression of PIP by SIP for the CNT ice nucleation is

illustrated in Figure 9. The ice nucleation is contributed from heterogeneous immersion,
deposition and contact ice nucleation. Among these mechanisms, the immersion freezing is
the dominant process in the single-layer mixed-phase clouds (Fig. 9a, b, c). The
contributions from deposition and contact ice nucleation to the total ice nucleation rate are
much smaller compared to immersion freezing. The immersion freezing rate is a function
of INPs in cloud droplets and temperature. CNT calculates the immersion freezing rate
based on cloud-borne BC and dust, the latter of which is the dominant INPs.

The immersion ice nucleation is weakened by a factor of 4.5 (Fig. 9a) after

considering SIP in the model due to lower number concentrations of INPs (Fig. 9d) and
cloud droplets (Fig. 9g). The cloud-borne dust number concentrations in the accumulation
(Fig. 9e) and coarse modes (Fig. 9f) are both decreased below ~750 hPa level,
corresponding to the reduction of INP number concentration and immersion ice nucleation
rate in CNT_SIP compared to the CNT experiment. Lower cloud-borne dust number
concentrations in the CNT_SIP experiment are caused by the reduction of cloud droplet
number concentrations (Fig. 9g) as a result of SIP. The SIP strongly enhances the accretion
of cloud water by snow (Fig. 9h) and the WBF process (Fig. 9i), leading to more
consumption of cloud water (Zhao and Liu, 2021). The ice crystals formed from SIP are
able to provide seeding for lower-level clouds when they sediment, further contributing
to the suppression of PIP. However, this effect may not be an important factor for the
suppression of PIP by SIP, considering that PIP occurs at higher levels relative to SIP in
the single-layer mixed-phase clouds (Figure 8).

The N12 and D15 schemes calculate the INP number concentrations based on the

interstitial aerosols (section 2.2). The mechanism for the suppression of PIP by SIP in the
case of the N12 ice nucleation is shown in Fig. S8: less cloud droplets and less available
interstitial aerosols (as a result of stronger wet deposition) with the introduction of SIP lead
to weaker PIP. The B53 and M92 schemes calculate the ice nucleation based on
temperature, supersaturation, and cloud droplet number concentration (section 2.2). Since
temperature is similar in these nudged simulations, the decreased cloud droplet number
concentration and ice supersaturation (due to the deposition of water vapor on more ice
crystals) with the introduction of SIP leads to weaker PIP in B53_SIP and M92_SIP.

On the other hand, ice nucleation can also compete with SIP. The ice nucleation

scheme with a larger ice nucleation rate (e.g., M92 versus CNT, Fig. 8a) is accompanied by
a smaller SIP rate (Fig. 8b). The peak SIP rate in M92_SIP is $\sim 10^{-14}$ kg kg$^{-1}$ s$^{-1}$, which is
about 10 times lower than that in CNT_SIP ($\sim 10^{-13}$ kg kg$^{-1}$ s$^{-1}$). This competition between
PIP and SIP is also revealed in the other ice nucleation schemes for the single-layer mixed-
phase cloud period (Fig. S5) and for the whole M-PACE period (Figs. S6 and S7). We note
that the largest PIP rate is M92, followed by B53, CNT, N12, and D15, while the SIP rate
is in the reversed order.

The mechanism for the suppression of SIP by PIP is illustrated in Figure 10. First, the

SIP rate is determined by three components, FR, IIC, and HM (Fig. 10a, b, c). The SIP rate
is dominated by IIC and FR. Second, the smaller FR rate in M92_SIP compared to that in
CNT_SIP (Fig. 10a) is a result of smaller rainwater mass mixing ratio (Fig. 10d), which is
caused by the strong M92 ice nucleation resulting in nearly complete glaciation of the
cloud in the M92_SIP experiment. Third, the IIC can be further subdivided into the non-
graupel-related IIC (Fig. 10e) and the graupel-related IIC (Fig. 10f), the latter of which
dominates the total IIC. A smaller graupel-related IIC rate (with the peak value of 2 $kg\ kg^{-1}$
$s^{-1}$) (Fig. 10f) in M92_SIP compared to CNT_SIP (with the peak value of 10 $kg\ kg^{-1}\ s^{-1}$) is
a result of smaller graupel mass mixing ratio in M92_SIP (with the peak value of 1.4 mg
$kg^{-1}$ in M92_SIP versus 5.2 $mg\ kg^{-1}$ in CNT_SIP) (Fig. 10g). As the graupel mass is
diagnosed from the cloud water mass, snow mass, and temperature, smaller mass mixing
ratios of cloud water (with the peak value of 8 versus 125 $mg\ kg^{-1}$ in Fig. 10h) and snow
(with the peak value of 1.4 versus 2.3 $mg\ kg^{-1}$ in Fig. 10i) in M92_SIP eventually lead to a
smaller graupel mass mixing ratio and a smaller graupel-related IIC rate. Similar results can
be found with the other ice nucleation schemes.

In summary, different from the PIP rate which is dependent on cloud-borne aerosols

and cloud droplets, the SIP rate is directly controlled by the precipitation particles, such as
rain, snow, and graupel. A stronger ice nucleation rate leads to more glaciation of mixed-
phase clouds in M92_SIP. As a consequence, less rainwater and graupel exist, leading to
lower SIP rate in the M92_SIP experiment compared to the CNT experiment.

## 448 5 Summary and conclusions

In this study, the relative importance of PIP through ice nucleation and SIP and their
interactions are investigated for the Arctic single-layer mixed-phase clouds observed
during M-PACE. To understand the interactions between PIP and SIP, five different ice
nucleation schemes (CNT, N12, D15, B53 and M92) are implemented in the model.
Model experiments with only ice nucleation and with both ice nucleation and SIP are
conducted. The CNT, N12, and D15 experiments without considering SIP show rather
constant ICNCs with cloud height, which are also one order of magnitude lower than the
observation. The SIP experiments based on the CNT, N12 and D15 ice nucleation schemes
(i.e., CNT_SIP, N12_SIP, and D15_SIP) reverse the vertical distribution pattern of ICNCs
by increasing the ICNCs in the lower portion of clouds. SIP also transforms ~30% of pure
liquid-phase clouds simulated in the CNT, N12, and D15 experiments into mixed-phase
clouds. In contrast, modeled clouds are totally ice phase instead of observed mixed-phase
in the B53 and M92 experiments. Since the cloud is already completely glaciated by the ice
nucleation with these ice nucleation schemes, adding the SIP processes has little impact on
the cloud phase in the B53_SIP and M92_SIP experiments. These findings highlight the
"foundation" effect of PIP on the cloud phase. We conclude that the model experiments
with both aerosol-aware ice nucleation schemes and SIP processes (i.e., CNT_SIP,
N12_SIP, and D15_SIP) yield the best agreement with observations in simulating the
Arctic single-layer mixed-phase clouds.
The relative importance of PIP and SIP is investigated in this study. We find that ice
nucleation contributes around 20% to the total ice production during M-PACE, with a
maximum value of 60% for the frontal clouds, and a minimum value of 7% for the single-
layer mixed-phase clouds in the CNT_SIP, N12_SIP, and D15_SIP experiments. The
B53_SIP and M92_SIP experiments may overestimate the contribution from PIP, which
contributes 65% and 80% to the total ice production, respectively averaged over the M-
PACE clouds.
In this study, for the first time, the interactions between PIP and SIP in the single-
layer mixed-phase clouds are investigated and possible mechanisms behind are discussed.
We find a clear suppression of PIP by SIP, and the ice nucleation rate is reduced when SIP
is introduced in the model. Ice crystals produced from SIP trigger a series of changes in
microphysical processes (e.g., WBF, riming), resulting in reduced number concentrations
of cloud droplets and cloud-borne dust aerosols. Less cloud-borne dust aerosols eventually
cause a weakening of the following ice nucleation (e.g., immersion freezing of cloud
droplets on dust). On the other hand, ice nucleation also competes with SIP. The ice
nucleation schemes with larger nucleation rates are accompanied by smaller SIP rates.
Different from the ice nucleation which depends on cloud water and aerosols, the SIP rate
is directly controlled by the precipitation particles. A stronger ice nucleation leads to more
glaciation of mixed-phase clouds, and as a consequence, less rain and graupel are formed,
leading to lower SIP rate.

We note that uncertainties still exist in the representations of ice nucleation and SIP in

the model. First, the diagnostic graupel approach still has a large uncertainty. A cloud
microphysical scheme with prognostic graupel (Gettelman et al., 2019b) or a "Single-Ice"
microphysical scheme (Morrison and Milbrandt, 2015; Zhao et al., 2017) will be needed to
further examine the impacts of graupel-related IIC. Second, modeled INP concentrations
may be significantly underestimated in the Arctic regions with the aerosol-aware CNT,
D15, and N12 ice nucleation schemes. This is owing to the model underestimation of long-
range transport of dust from lower latitudes (Shi and Liu, 2019) as well as the model
missing of high-latitude local dust (Shi et al., 2021) and marine biogenic aerosols in the
Arctic regions (Zhao et al., 2021b). Our future work will focus on representing the high
latitude dust and biological aerosol emissions for better INP simulations in the model as
well as improving the parameterization of SIP processes. More observation data are needed
to identify the frequencies and conditions of SIP occurrence in cold clouds and its
contribution to total ice formation so that the impact of SIP can be better quantified by the
models.

**Competing interests:** The authors declare that they have no conflict of interest.

**Data availability:** The Community Earth System Model version 2 (CESM) source code is
freely available at http://www. cesm.ucar.edu/models/cesm2 (Danabasoglu et al., 2020;
last access: 3 July 2021). The SIP source code and model datasets are archived at the NCAR
Cheyenne supercomputer and are available upon request. The measured LWP and IWP
datasets of M-PACE campaign are obtained from the Atmospheric Radiation Measurement
(ARM) user facility, US Department of Energy Office of Science, available at
https://www.arm.gov/research/campaigns/nsa2004arcticcld (McFarquhar et al., 2007; last
access: 3 July 2021).

**Author contributions**: XZ and XL conceptualized the analysis,carried out the simulations,
performed the analysis, and wrote the manuscript. XL was involved with obtaining the
project grant and supervised the study.

**Acknowledgment:** We thank Vaughan T. J. Phillips, and Sachin Patade for helpful
discussions. We thank Meng Zhang for helpful discussions, especially on processing the
observation data. The authors would also like to acknowledge the use of computational
resources for conducting the model simulations (ark:/85065/d7wd3xhc) at the NCAR-
Wyoming Supercomputing Center provided by the NSF and the State of Wyoming and
supported by NCAR's Computational and Information Systems Laboratory.

**Financial support:** This research was supported by the DOE Atmospheric System
Research (ASR) Program (grants DE-SC0020510 and DE-SC0021211).

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

Table 1. List of model experiments.

| Experiment | Secondary Ice Production | Ice Nucleation |
|:---:|:---:|:---:|
| **CNT** | HM | Default model with CNT ice nucleation |
| **N12** | HM | Niemand et al. (2012) ice nucleation |
| **D15** | HM | DeMott et al. (2015) ice nucleation |
| **B53** | HM | Bigg (1953) ice nucleation |
| **M92** | HM | Meyers et al. (1992) ice nucleation |
| **CNT_SIP** | HM, FR, IIC | CNT ice nucleation |
| **N12_SIP** | HM, FR, IIC | Niemand et al. (2012) ice nucleation |
| **D15_SIP** | HM, FR, IIC | DeMott et al. (2015) ice nucleation |
| **B53_SIP** | HM, FR, IIC | Bigg (1953) ice nucleation |
| **M92_SIP** | HM, FR, IIC | Meyers et al. (1992) ice nucleation |


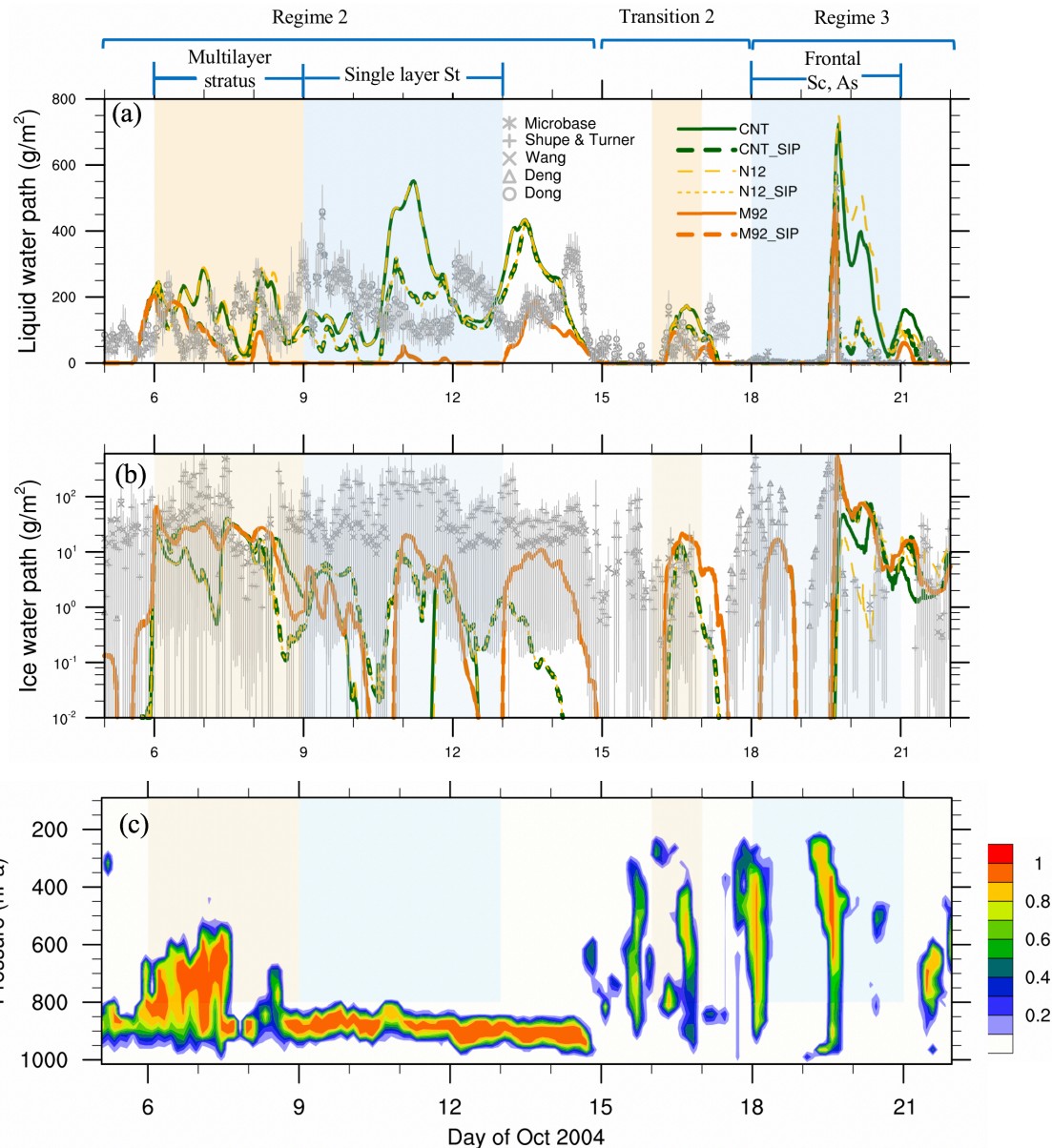


Figure 1. Temporal evolution of (a) LWP and (b) IWP from remote sensing retrievals
(symbols) and CNT, CNT_SIP, N12, N12_SIP, M92, and M92_SIP experiments (lines);
(c) vertical distribution of observed cloud fraction. The light orange shadings show the
multilayer stratus and transition periods; light blue shadings show the single-layer stratus
and frontal clouds periods. Vertical gray lines represent the standard deviations of retrieval
data. Note that N12 (N12_SIP) coincides with CNT (CNT_SIP) during the single-layer
stratus cloud period.

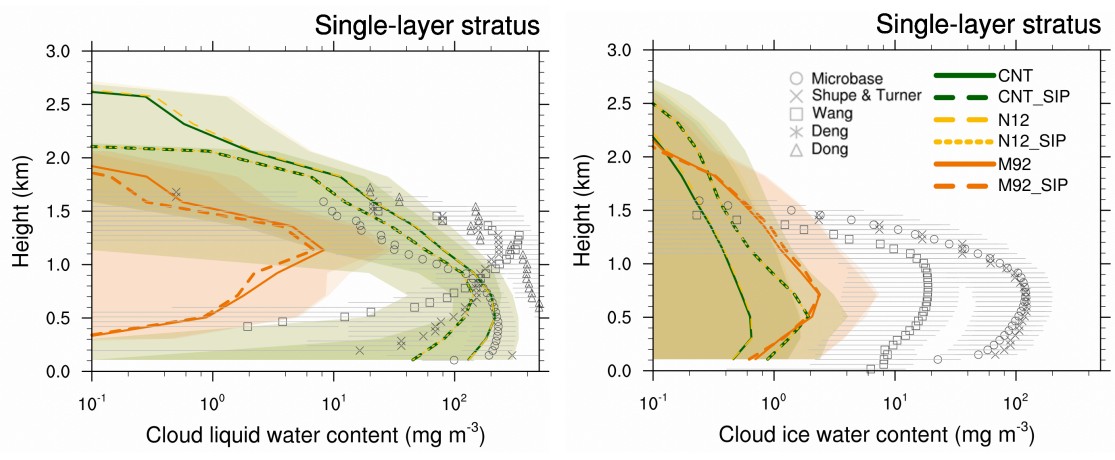


Figure 2. Vertical profiles of LWC (left) and IWC (right) during the single-layer mixed-
phase cloud period (October 9-12) from CNT, CNT_SIP, N12, N12_SIP, M92, and
M92_SIP experiments and from remote sensing retrievals (symbols). Horizontal gray lines
represent standard deviations of retrieval data, and colored shadings are standard
deviations of model data. Note that N12 (N12_SIP) coincides with CNT (CNT_SIP)
during the single layer stratus cloud period.

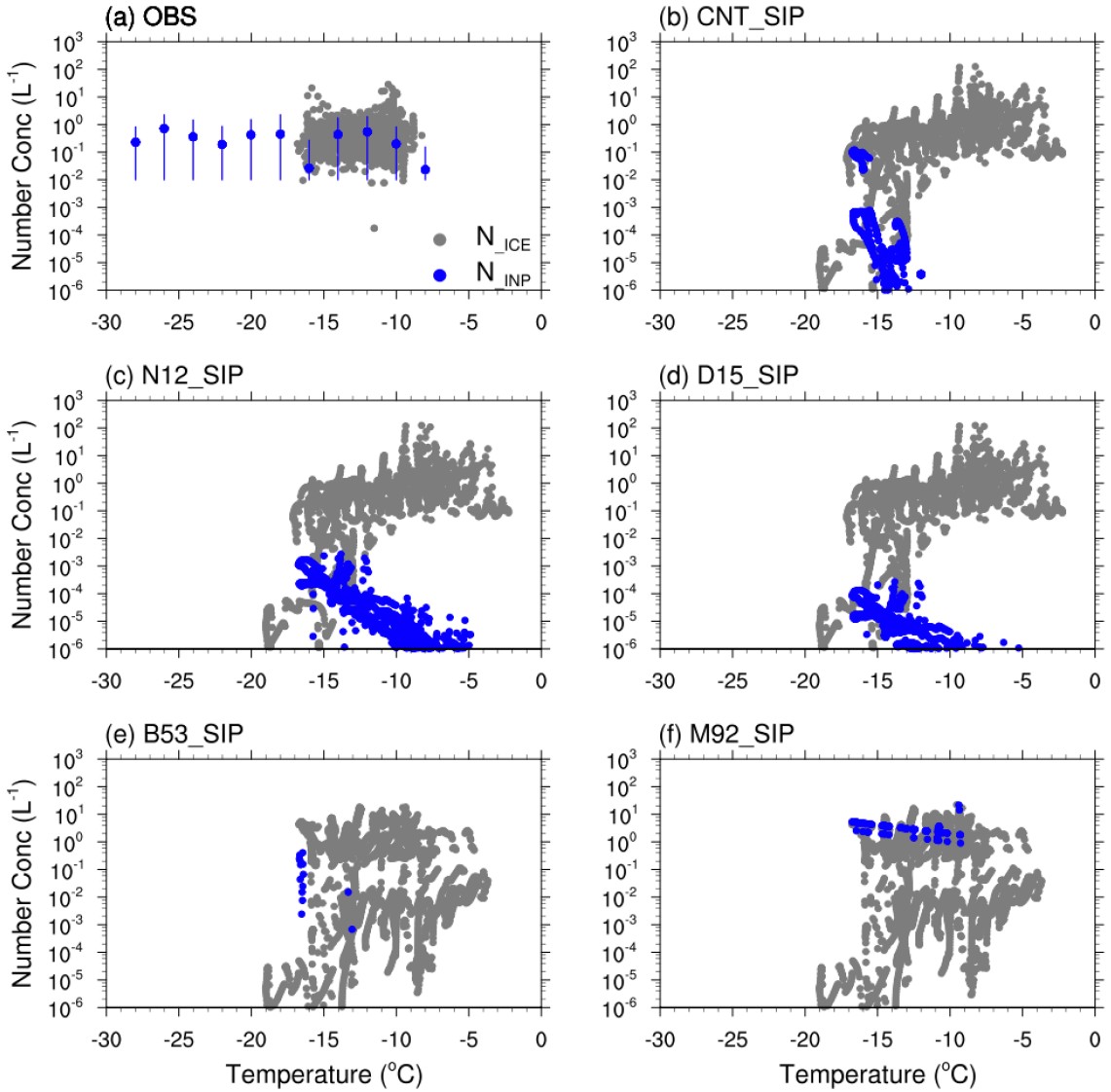


Figure 3. Comparison between INP (blue dots, in unit of $L^{-1}$) and ice crystal number

concentrations (gray dots, in unit of $L^{-1}$) from (a) observations, (b) CNT_SIP, (c)

N12_SIP, (d) D15_SIP, (e) B53_SIP, and (f) M92_SIP experiments. Modeled ice number

concentrations include ice crystals of all sizes, since the purpose of this figure is to

compare INP number concentrations with ice crystal number concentrations. To account

for the anti-shattering tip effect, only ice particles with diameters larger than 100 μm

from observations are included in Fig. 3a, and a correction factor of 1/4 is also applied to

the measured ice crystal number concentrations based on Jackson et al. (2014) and

Jackson and McFarquhar (2014). The purpose of this figure is to examine the relative

importance between primary ice nucleation and SIP by comparing INP and ice crystal
number concentrations. Therefore, all ice sizes are included in the simulation results.

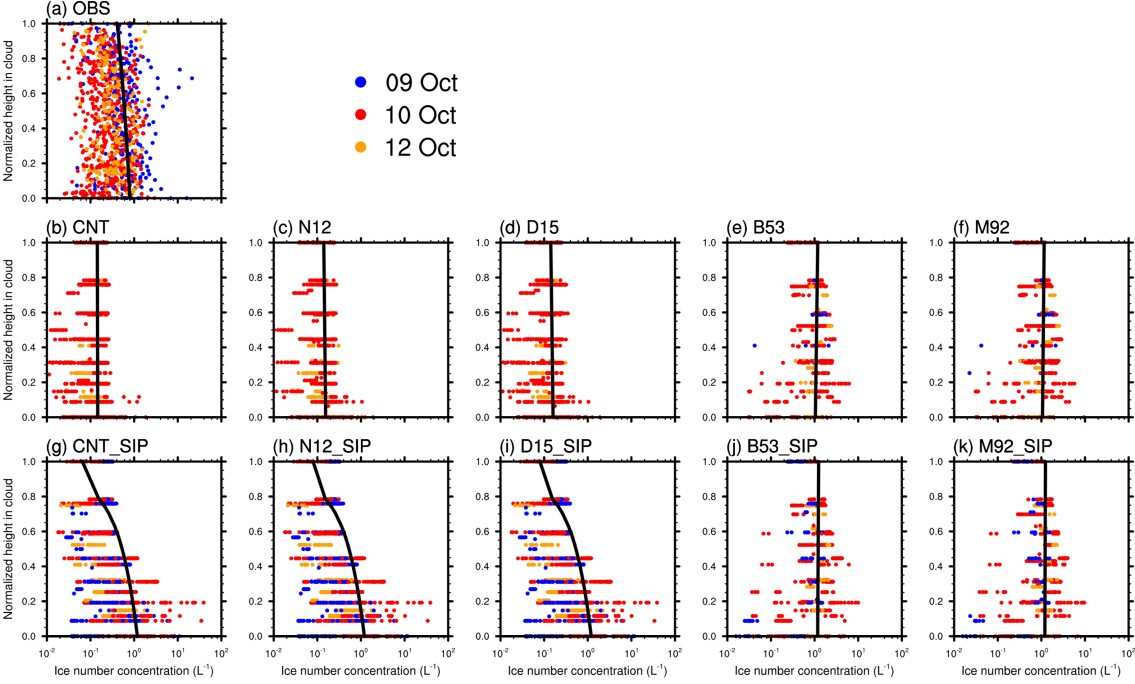


Figure 4. Ice crystal number concentrations as a function of normalized cloud height (i.e.,
0 for cloud base and 1 for cloud top) from (a) observation, (b) CNT, (c) N12, (d) D15, (e)
B53, (f) M92, (g) CNT_SIP, (h) N12_SIP, (i) D15_SIP, (j) B53_SIP, and (k) M92_SIP
experiments. Black solid lines show the linear regression between ice number
concentration and height. Only ice particles with diameters larger than 100 μm from
simulations and observations are included in the comparison. To account for the anti-
shattering tip effect, a correction factor of 1/4 is applied to the measured ice number
concentrations based on Jackson et al. (2014) and Jackson and McFarquhar (2014). The
cloud base and cloud top used for (a) are provided from in situ observations (McFarquhar
et al., 2007), and those used for the model analyses are derived by searching the model
layers from the model top to the bottom with modeled total cloud water LWC+IWC $>10^{-}$
$^{6}$ kg kg$^{-1}$.

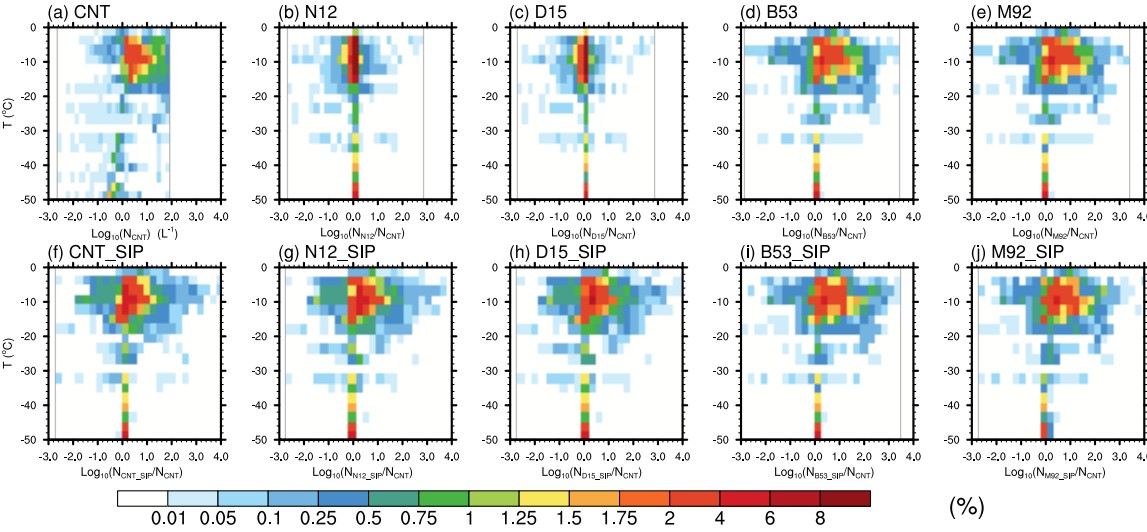

Figure 5. Bivariate joint probability density functions (PDF) in terms of both temperature and (a) ice crystal number concentration ($L^{-1}$) from the CNT experiment, and (b)-(j) in terms of both temperature and enhancement ratio of ice crystal number concentration from the respective experiment to that from the CNT experiment. A logarithmic scale is used for the x-axis.

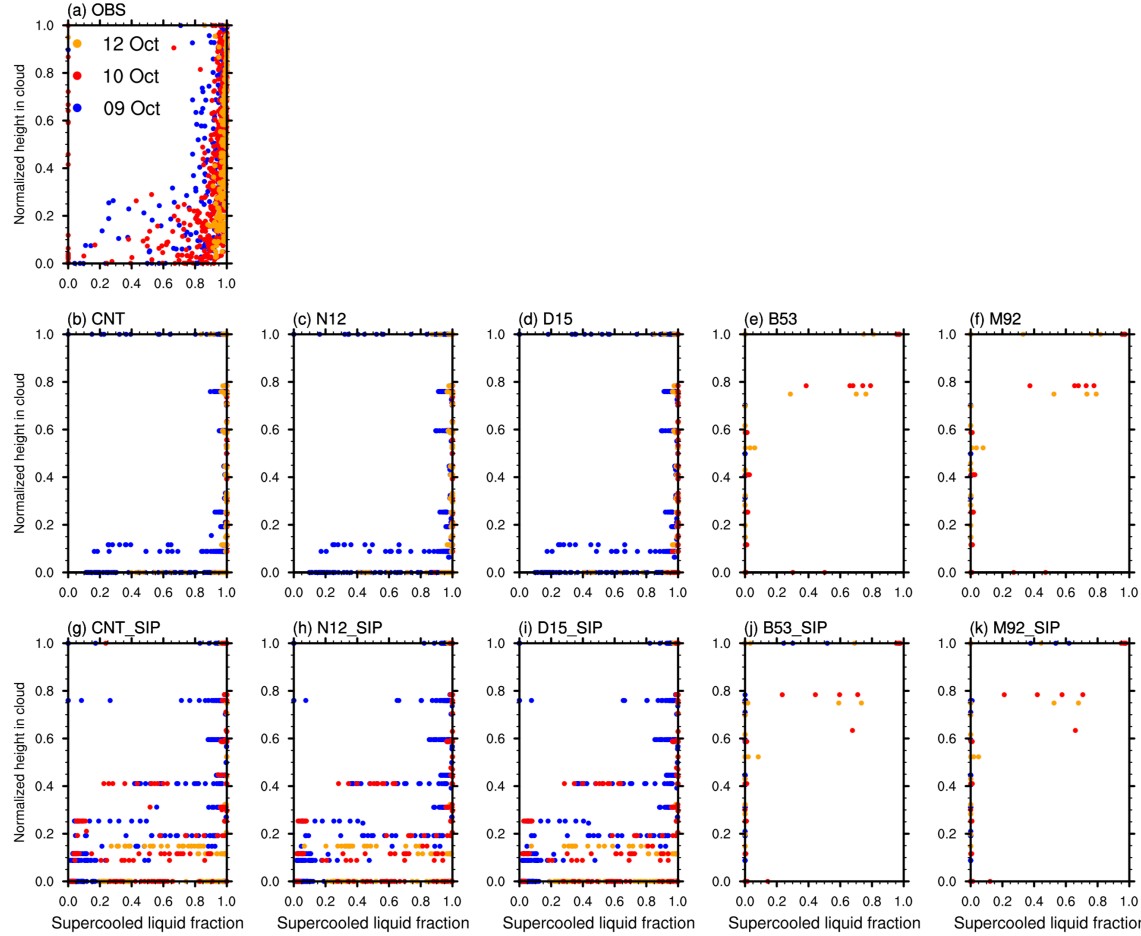

Figure 6. Supercooled liquid fraction (defined as LWC/(LWC + IWC)) as a function of

normalized cloud height (i.e., 0 for cloud base and 1 for cloud top) from observations and

model experiments. The cloud base and cloud top used for (a) are provided from in situ

observations (McFarquhar et al., 2007), and those used for the model analyses are derived

by searching the model layers from the model top to the bottom with modeled total cloud

water LWC+IWC $> 10^{-6}$ kg kg$^{-1}$.

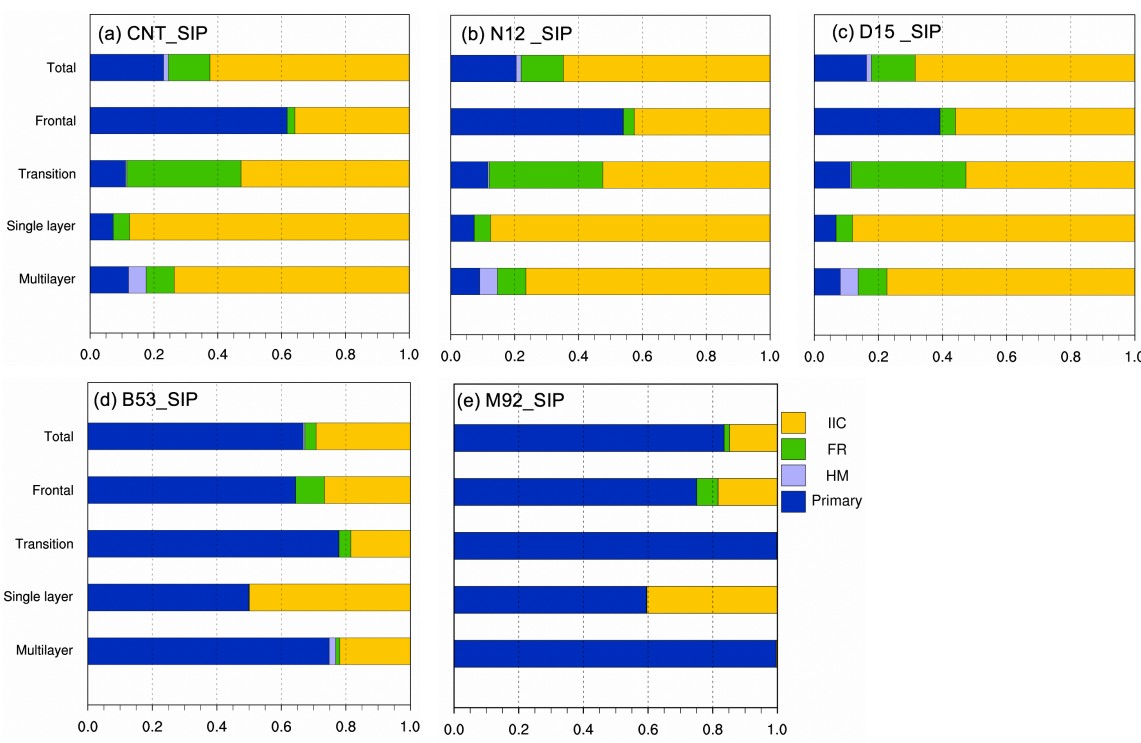

Figure 7. Stacked bar charts of relative contributions from ice nucleation and secondary ice production to the total ice production rate from (a) CNT_SIP, (b) N12_SIP, (c) D15_SIP, (d) B53_SIP, and (e) M92_SIP experiments averaged over different time periods of M-PACE. The secondary ice production includes ice-ice collisional breakup (IIC), rain droplet fragmentation (FR), and Hallett–Mossop (HM) process.

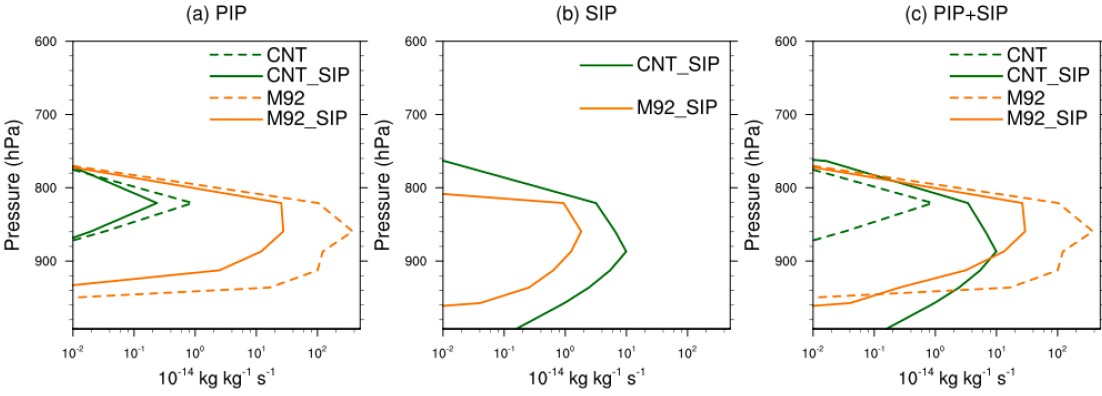


Figure 8. Vertical profiles of (a) primary ice production rate (unit: kg kg$^{-1}$ s$^{-1}$), (b)
secondary ice production rate (unit: kg kg$^{-1}$ s$^{-1}$), and (c) primary plus secondary ice
production rate (unit: kg kg$^{-1}$ s$^{-1}$) from CNT, CNT_SIP, M92, and M92_SIP model
experiments averaged over the single-layer mixed-phase cloud period. Ice production rates
are grid-box means.

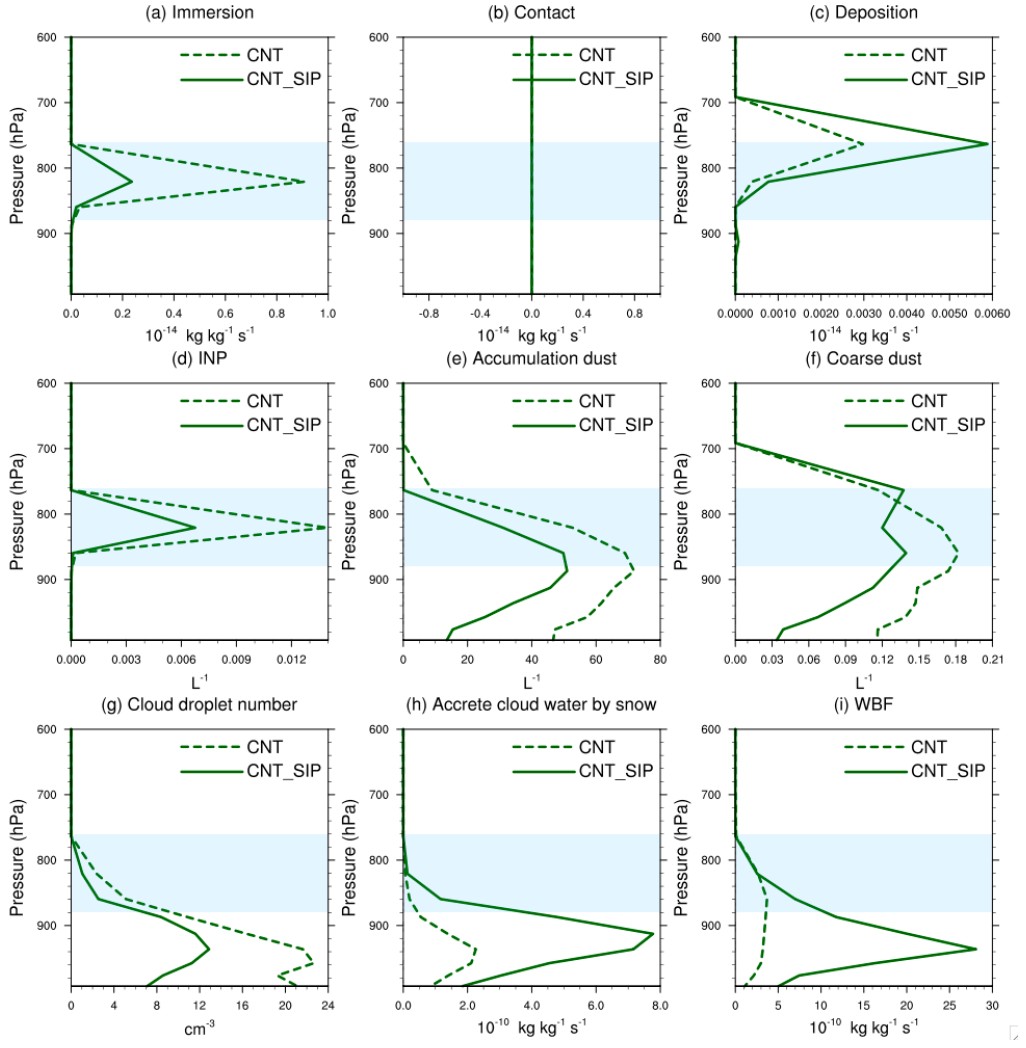


Figure 9. Vertical profiles of (a) ice production rate (unit: kg kg⁻¹ s⁻¹) from immersion
freezing of cloud water, (b) ice production rate (unit: kg kg⁻¹ s⁻¹) from contact freezing of
cloud water, (c) ice production rate (unit: kg kg⁻¹ s⁻¹) from homogeneous and
heterogeneous deposition nucleation, (d) immersion freezing INP number concentration,
(e) cloud-borne dust number in the accumulation mode, (f) cloud-borne dust number in
the coarse mode, (g) cloud droplet number concentration, (h) accretion rate of cloud
droplets by snow, and (i) WBF process rate from CNT and CNT_SIP experiments
averaged over the single-layer mixed-phase cloud period. Light blue shadings indicate the
ice nucleation regime. Ice production rates are grid-box means.

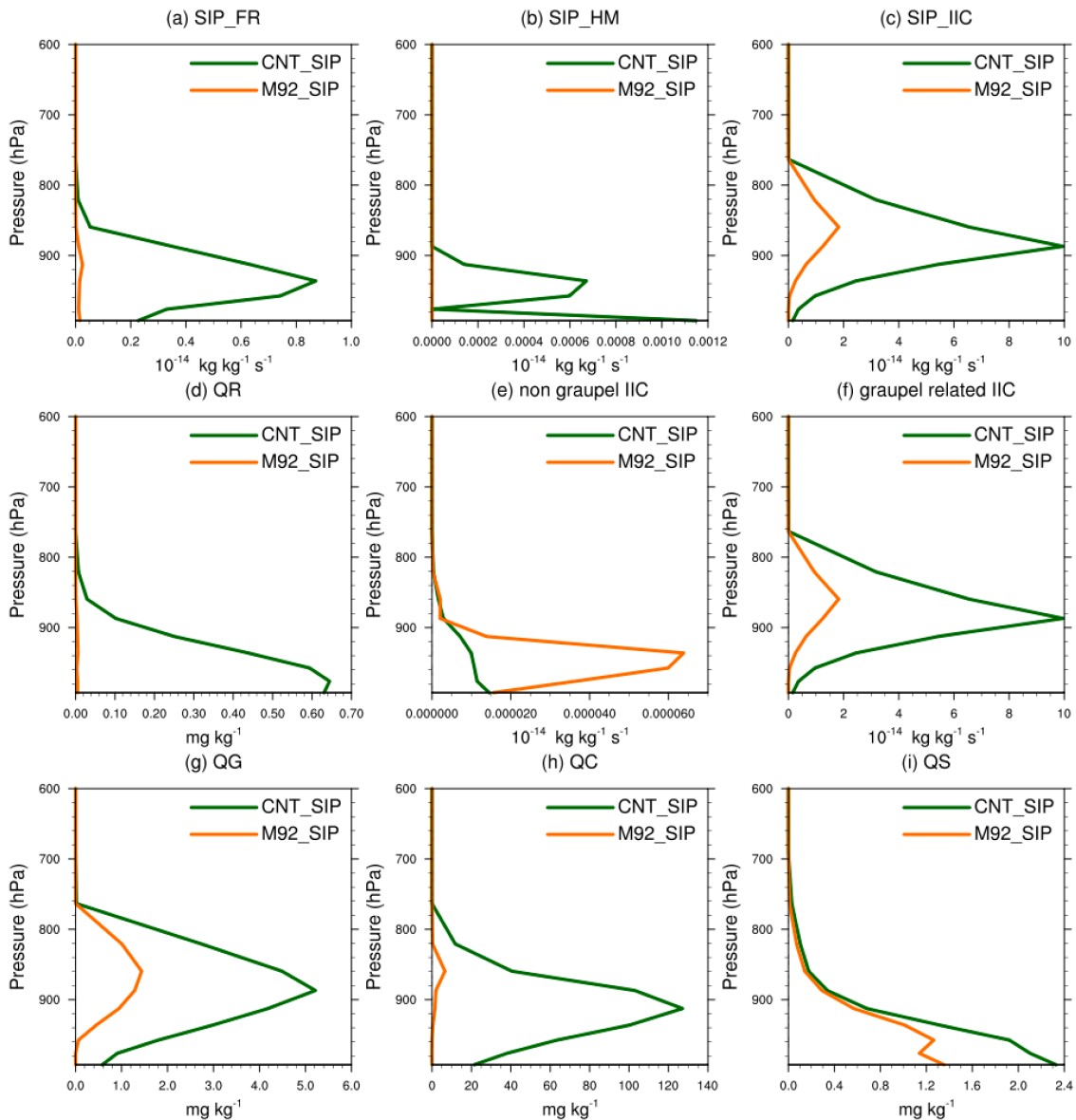


Figure 10. Vertical profiles of (a) rain droplet shattering rate during freezing (FR), (b)

rime splintering rate (HM), (c) ice-ice collision fragmentation rate (IIC), (d) rain water

mixing ratio (Qr, in unit of mg kg$^{-1}$), (e) non graupel related ice-ice collision

fragmentation rate, (f) graupel related ice-ice collision fragmentation rate, (g) graupel

mass mixing ratio (Qg, in unit of mg kg$^{-1}$), (h) cloud water mass mixing ratio (Qc, in unit

of mg kg$^{-1}$), and (i) snow mass mixing ratio (Qs, in unit of mg kg$^{-1}$) from the CNT_SIP

and M92_SIP experiments.

833