# Peer review of "Primary and Secondary Ice Production: Interactions and Their Relative Importance"

_Atmospheric Chemistry and Physics, 2021_

## Author Response (AR2)

**Response to Reviewer 1**

We thank the anonymous reviewer for his/her careful reading and constructive review of our paper. Our detailed responses to the comments follow. Reviewer's comments are in blue color, our responses are in black color, and our corresponding revisions in the manuscript are in red color.

Review of "Relative importance and interactions of primary and secondary ice production in the Arctic mixed-phase clouds" by Zhao and Liu in ACPD, 2021.

In this work, the authors contrasted several parameterizations of primary ice production (PIP), combined with a new set of parameterizations of secondary ice production (SIP) in the NCAR CESM2/CAM6 model. The model simulations are compared with observations from the DOE M-PACE campaign. The scientific questions include: What are the impacts of SIP parameterizations on the simulation results? What are the effects of SIP on PIP? How does the PIP process influence SIP? As the authors mentioned, the interactions of SIP and PIP have not been carefully examined before, and the mechanisms of how they affect each other are still unclear.

Overall, this is a well-written manuscript. It is very easy to follow the simulation experimental design since the logics are very clear and straightforward. The reviewer recommends that the paper be accepted after a minor revision on the following points.

**Reply:** We thank the reviewer for the encouraging comments. We have revised our manuscript following your comments regarding the observation data and clarified the text to improve the quality of our paper.

**Main comments:**

1. About the comparison of ice crystal number concentration (ICNC) between observations and simulations, the observations are restricted to > 100 micron, while the simulations use the entire size range from zero to infinity. Since ICNC is dominated by smaller ice particles, the simulations may overestimate ICNC when a wider size range is used. The reviewer suggests a revision on the simulation dataset to delete ice particles < 100 micron. In addition, a scaling factor of 1/4 is applied to the observations due to potential ice shattering effect. But as the author mentioned, previous studies showed that the scaling factor may be around 1 to 1/4.5. Thus, using 1/4 seems to provide a lower end of ICNC from observations. If the authors apply another scaling factor, such as 1/2, how will it change the result? Some discussions on this sensitivity test can be added.

**Reply:** We thank the reviewer for the suggestion. Following the reviewer's suggestion, we have replotted Figure 3 in which only ice particles larger than 100 microns are used, shown as Figure R1 below. The purpose of Figure 3 is to examine the relative importance between primary nucleation and SIP by comparing INP and ice number concentrations. The idea is, INPs represent the primary nucleated ice, and the difference between INP and total ice number concentrations reflects the impact of SIP. Here, we are not comparing the ice crystal number concentrations between observations and simulations. Therefore, we used all sizes of ice crystals from the simulations in Figure 3 in the manuscript. We have added a note in the Figure 3 caption: "The purpose of this figure is to examine the relative importance between primary ice nucleation and SIP by comparing INP and ice crystal number concentrations. Therefore, all ice sizes are included in the simulation results".

We have Figure 4 in the manuscript which is specifically aimed at comparing the simulated and the observed ice number concentrations. Figure 4 in the manuscript already uses the simulated ice larger than 100 microns.

We agree with the reviewer that discussions on the observation sensitivity to a different scaling factor is necessary. We conducted a sensitivity test with a scaling factor of 1/2 as the reviewer suggested, to the observed ICNC, as shown in supplementary Figure S3 (attached below as Figure R2). We have added some discussions about this sensitivity test in the main text:

"A different scaling factor of 1/2 is applied to the observed ICNCs, which increases the observed ICNCs by a factor of 2 (Figure S3). The underestimation of ICNCs by the model experiments with only ice nucleation (CNT, N12 and D15) is even worse and our conclusion regarding model and observation comparison of ICNCs is not changed."

[Figure]

Figure R1. Same as Figure 3 but only shows ice particles with diameters larger than 100 μm from all the simulations.

[Figure]

Figure R2. Same as Figure 3a and Figure 4a, but applied a correction factor of 1/2 to the measured ice crystal number concentrations for Figure 3a (left) and Figure 4a (right).

2. Another main comment is about the mechanism used to explain how introduction of SIP leads to weaker PIP. The authors described this mechanism around line 339 - 346, that is, "Since temperature and supersaturation are similar in these nudged simulations, the decreased cloud droplet number concentration with the introduction of SIP leads to weaker PIP in B53_SIP and M92_SIP". Can the authors clarify which variables in the SCAM simulation are nudged, such as temperature, U and V wind? Is the specific humidity nudged as well? The reviewer tries to understand why ice supersaturation is similar between the two simulations. If there are more ice crystals produced by SIP, these ice crystals could
provide more deposition of water vapor to ice phase, and thereby relaxing ice
supersaturation back to ice saturation faster. Then it could lead to a suppression of PIP
when ice supersaturation frequency and/or magnitude is reduced, since PIP requires a
certain magnitude of ice supersaturation to occur. Also, are the ice crystals formed from
SIP able to provide seeding for lower levels when they sediment? Can the seeding lead to
suppression of PIP?

**Reply:** We thank the reviewer for the great comment. In the SCAM simulations, wind (U
and V) and temperature are nudged, while the specific humidity is not. We plotted the
vertical distribution of the relative humidity with respect to ice (RHice) (Figure R3 below),
and indeed the RHice is lower in M92_SIP than in M92, consistent with the reviewer's
comment that more ice crystals produced by SIP should lead to more deposition of water
vapor and reduce RHice in the SIP simulations. We also agree with the reviewer that this
will further suppress the PIP in the SIP simulations.

We modified the sentence in the revised manuscript as: "Since temperature is similar in
these nudged simulations, the decreased cloud droplet number concentration and ice
supersaturation (due to the deposition of water vapor on more ice crystals) with the
introduction of SIP leads to weaker PIP in B53_SIP and M92_SIP".

[Figure]

Figure R3. Relative humidity with respect to ice (RHice) from M92 and M92_SIP
experiments averaged over the single-layer mixed-phase cloud period.

Yes, ice crystals formed from SIP are able to provide seeding for lower level clouds when
they sediment. The seeding can lead to suppression of PIP. We have added some sentences
in the revised manuscript to discuss the contribution of the seeding effect. However, this
effect may not be an important factor in the single-layer mixed-phase clouds, since PIP

occurs in relatively higher cloud levels compared with SIP (Figure 8a and b), and low-level PIP may not contribute significantly to the ice formation. Ice seeding from multi-layer clouds is not important in this single-layer cloud period.

We have added in the revised manuscript: "The ice crystals formed from SIP are able to provide seeding for lower-level clouds when they sediment, further contributing to the suppression of PIP. However, this effect may not be an important factor for the suppression of PIP by SIP, considering that PIP occurs at higher levels relative to SIP in the single-layer mixed-phase clouds (Figure 8)."

3. Following the previous comments on Figure S7, some parts of this figure do not make sense to the reviewer. For example, accumulation mode dust decreases at 880 – 1000 hPa, but increases at 880 – 700 hPa in N12_SIP compared with N12. Why does the accumulation mode dust increase at 880 – 700 hPa in N12_SIP, if the mechanism of SIP is to increase wet deposition (line 341)? In addition, the accumulation mode deposition in panel (e) only significantly increases in N12_SIP near the surface around 980 – 1000 hPa. This pressure level does not match with the location of changes seen in panel (d), and the increasing deposition doesn't explain the increase of accumulation mode dust at 880 – 700 hPa as mentioned above. The change of coarse mode deposition also doesn't match with the vertical locations of changes seen in coarse mode dust. Can the authors explain this figure a bit more?
Some minor comments on Figure S7, the (d) panel x axis label is out of bound on the page. Also some x axes are suggested to use the same range for an easier comparison. For example, c, d, and e can use the same scale and unit; f and g can use the same scale.

**Reply:** We thank the reviewer for the helpful comment and suggestion. We have taken a more careful look at the changes of accumulation mode dust number concentration in Figure S7d, and found that the changes are actually neglectable (~1%) compared to its absolute concentrations (0.3 versus 30 L$^{-1}$). Also, the accumulation mode dust contributes much less to primary ice nucleation than the coarse mode dust. Therefore, we have removed the changes in the accumulation mode dust number concentration (Figure S7d) and in the deposition rate of accumulation mode dust (Figure S7f) in the revised manuscript. We agree with the reviewer's comment that the change of coarse mode dust deposition does not match with the vertical locations of changes in coarse mode dust, since the changes of aerosols are influenced by other processes, such as horizontal and vertical advection, in addition to wet/dry deposition. We plot only wet deposition rate of interstitial coarse mode dust in Figure S7g (now Figure S8e). As can be seen, the stronger (more negative as a sink term) wet scavenging leads to less coarse mode dust at 850-1000 hPa. Changes of coarse mode dust at 820-700 hPa (which is above the cloud layers) are mainly due to other processes such as aerosol transport, and not to cloud processes. The purpose of Figure S7

(now Figure S8) is to explain that the weaker primary ice nucleation is caused by lower cloud droplet number and less INP. Less INP is mainly due to less coarse mode dust, and the stronger wet deposition can explain the decreases of coarse mode dust at 850-1000 hPa.

Following the reviewer's comment, we have used the same unit and scale for the x-axes of

INP and coarse mode dust number concentrations in the revised Figure S8. The original panels (d and f) for accumulation mode dust are removed. Revised Figure S8 looks:

[Figure]

Figure R4. Vertical profiles of differences of (a) cloud droplet number concentration, (b)

ice production rate from immersion freezing of cloud droplets, (c) immersion freezing INP

number concentration, (d) interstitial dust number concentration in the coarse mode, and (e) wet deposition rate of interstitial coarse mode dust between the N12_SIP and N12

experiments.

4. In several analyses, the authors use relative altitude to the cloud layer, that is, 0 refers to
cloud base and 1 refers to cloud top. There is no discussion about how this relative altitude
is derived. Is it derived based on ground-based observations or in-situ observations? Please
clarify.

**Reply:** The relative altitude (and the associated cloud base and cloud top) used in the
observational analysis (in Figures 4 and 6) are provided from McFarquhar et al. (2007),
which are derived based on in situ observations. For our model analysis, we assume that
clouds exist when the total cloud water LWC+IWC $>10^{-6}$ kg kg$^{-1}$. From the model top to
the bottom, the first model layer with LWC+IWC $>10^{-6}$ kg kg$^{-1}$ is assigned as the cloud
top, and similarly the last model layer with LWC+IWC $>10^{-6}$ kg kg$^{-1}$ is assigned as the
cloud base.

We have added the following note in the captions of Figures 4 and 6:
"The cloud base and cloud top used for (a) are provided from in situ observations
(McFarquhar et al., 2007), and those used for the model analyses are derived by searching
the model layers from the model top to the bottom with modeled total cloud water
LWC+IWC $>10^{-6}$ kg kg$^{-1}$."

Minor comments:
1. Several simulations, CNT, N12 and D15, as well as CNT_SIP, N12_SIP and D15_SIP,
provide similar results to each other. Can the authors provide some explanations why these
three PIP parameterizations provide very similar results? Is it because they were derived
based on similar observation data?

**Reply:** We thank the reviewer for the comment. Although these three parameterization
schemes differ in details about temperature and aerosol dependences, CNT, N12, and D15
predict much lower INP concentrations for the M-PACE single-layer clouds compared with
the B53 and M92 schemes. With low INP concentrations, modeled clouds are
overwhelmingly dominated by liquid-phase. Therefore, it is not surprising to see the overall
similar cloud features among the CNT, N12, and D15 simulations. In contrast, B53 and
M92 which are only dependent on temperature and not limited by aerosols predict much
higher INP concentrations. With these high INP concentrations, modeled clouds with the
B53 and M92 schemes are dominated by ice-phase.

We have added a note when we discuss about Figure 2 in the revised manuscript:
"Although these schemes differ in details about temperature and aerosol dependences
(Figure 3), CNT, N12, and D15 predict much lower INP concentrations during M-PACE

than those from the B53 and M92 schemes. With these low INP concentrations, the single-layer clouds modeled with the CNT, N12 and D15 schemes have similar cloud states (e.g., dominated by liquid-phase) (Figures 1 and 2). In contrast, B53 and M92 which are only dependent on temperature and not limited by aerosols predict much higher INP concentrations. With these high INP concentrations, modeled clouds with the B53 and M92 schemes are dominated by ice-phase."

2. Some of the analyses and figures are based on ground-based remote sensing observations (such as Figure 1) while the other ones are based on in-situ aircraft observations. It would be beneficial to clarify in Section 3, such as line 182, which type of observations is used in a specific figure or analysis.

**Reply:** We thank the reviewer for the suggestion. We have added: "The ice water path (IWP) and liquid water path (LWP) are based on ground-based remote sensing observations provided by Zhao et al. (2012) with uncertainties within one order of magnitude (Dong and Mace, 2003; Shupe et al., 2005; Deng and Mace, 2006; Turner et al., 2007; Wang, 2007; Khanal and Wang, 2015). The INP concentrations are based on in-situ observations by a CFDC on board an aircraft (Prenni et al., 2007). The ICNCs and cloud phase are based on in-situ observations and provided by McFarquhar et al. (2007)." in Section 3 to clarify the types of observation data used in the analyses.

3. Please clarify how the variables related to "rate" are defined in the model. For example, is the variable ice production rate describing the amount of ice crystals (in kg) being produced in every unit mass of dry air (in kg) per second in the entire grid box, or only in the in-cloud section of the grid box?

**Reply:** We thank the reviewer for the comment. In the model, the unit of ice production rates is kg (ice crystals)/(kg dry air)/s, and all the ice production rates are grid box mean values. We have added the unit and a note to the captions of Figures 8 and 9.

4. Figure 2, is it possible to add sub-panels of observations to compare with the model results?

**Reply:** We thank the reviewer for the suggestion. We have added the observation data (including standard deviations) in Figure 2, and the revised figure looks:

[Figure]

Figure R5. Vertical profiles of LWC (left) and IWC (right) during the single-layer mixed-
phase cloud period (October 9-12) from CNT, CNT_SIP, N12, N12_SIP, M92, and
M92_SIP experiments and from remote sensing retrievals (symbols). Horizontal gray lines
represent standard deviations of retrieval data, and colored shadings represent standard
deviations of model data. Note that N12 (N12_SIP) coincides with CNT (CNT_SIP) during
the single layer stratus cloud period.

5. Figure 3, since the INP concentrations in CNT, N12 and D15 are significantly lower
than the observations, can the authors apply a scaling factor to INPs in these
parameterizations to match with the observations better, and see how the results change?
Also, the reviewer wonders why with such low INP concentrations, these parameterizations
are able to produce quite a similar amount of ICNC compared with observations?

**Reply:** We thank the reviewer for the good suggestion. Applying a scaling factor to INPs
means changing the INP parameterizations. Instead, we undertook another approach by
increasing dust aerosol concentrations used in the INP parameterizations, which will also
result in more INPs. We have conducted a sensitivity test using the CNT scheme with
increased dust concentrations by 100 times. This simulation shows overall similar cloud
properties, but the relative contribution of primary ice nucleation to total ice production is
increased by a factor of ~2 during M-PACE. We have added a discussion on this in Section
4.2 in the revised manuscript as: "Since the INP number concentrations in CNT, N12 and
D15 are significantly lower than the observations (Figure 3), a sensitivity test using the CNT
scheme with increased dust concentrations by 100 times shows overall similar cloud
properties. However, the relative contribution of primary ice nucleation to total ice production is increased by a factor of ~2 to 30% averaged for all the cloud types and to 20%
for the single-layer mixed-phase clouds."

For the reviewer's question: "why such low INP concentrations produce quite a similar
amount of ICNC compared with observations?" The ICNCs in CNT, N12, and D15
experiments are actually 1-2 orders of magnitude lower than observed ICNCs as shown in
Figure 4, not at a similar amount. The simulations with SIP (using CNT, N12, and D15)
produce similar amounts of ICNCs at the lower portion of clouds compared with
observations.

6. Figure 5, please clarify how the normalization was calculated in this figure. It seems that
the PDF is calculated by the number of samples of each bin divided by the total number of
samples in each temperature bin (the sum of % in each temperature range equals one),
instead of divided by the total number of samples of the entire temperature range. Is that
correct? The reviewer wonders how this figure will change, if the latter type of
normalization is also provided (i.e., the sum of % in all bins equals one).
**Reply:** We thank the reviewer for the question. We are sorry for the confusion. Figure 5
shows the probability of occurrence defined in terms of both temperature and ice number
concentration for Figure 5a (or enhancement ratio for Figure 5b-j), which means PDF is
calculated by the number of samples of each bin divided by the total number of samples of
the entire temperature range (i.e., the sum of % in all bins equals one). We have revised
the caption of Figure 5 as: "Figure 5. Bivariate joint probability density functions (PDF) in
terms of both temperature and (a) ice crystal number concentration ($L^{-1}$) from the CNT
experiment, and (b)-(j) in terms of both temperature and enhancement ratio of ice crystal
number concentration from the respective experiment to that from the CNT experiment. A
logarithmic scale is used for the x-axis."

7. Figure 9, for the accretion rate of cloud water by snow, does cloud water include both
cloud droplets and rain? Some minor revisions on the sub-title of g and h are recommended.
For example, h can be "Droplet number" instead of "Cloud number", and h can be "Accrete
water by snow".
**Reply:** We thank the reviewer for the question and suggestion. Figure 9h is the accretion
rate of cloud droplets by snow, and the cloud water does not include rain. We have revised
the sub-titles of Figure 9g and h as you suggested to "Cloud droplet number" and "Accrete
cloud water by snow", respectively. The revised Figure 9 looks:

[Figure]

Figure R6. Vertical profiles of (a) ice production rate (unit: kg kg$^{-1}$ s$^{-1}$) from immersion freezing of cloud water, (b) ice production rate (unit: kg kg$^{-1}$ s$^{-1}$) from contact freezing of cloud water, (c) ice production rate (unit: kg kg$^{-1}$ s$^{-1}$) from homogeneous and heterogeneous deposition nucleation, (d) immersion freezing INP number concentration, (e) cloud-borne dust number in the accumulation mode, (f) cloud-borne dust number in the coarse mode, (g) cloud droplet number concentration, (h) accretion rate of cloud droplets by snow, and (i) WBF process rate from CNT and CNT_SIP experiments averaged over the single-layer mixed-phase cloud period. Light blue shadings indicate the ice nucleation regime. Ice production rates are grid-box means.

**Response to Reviewer 2**

We thank the anonymous reviewer for his/her careful reading and constructive review of our paper. Our detailed responses to the comments follow. Reviewer's comments are in blue color, our responses are in black color, and our corresponding revisions in the manuscript are in red color.

Review of Manuscript # acp-2021-686 in ACPD: "Relative importance and interactions of primary and secondary ice production in the Arctic mixed-phase clouds" by Zhao and Liu.
**General comments:**
The authors examined five different ice nucleation schemes and secondary ice production (SIP) processes in the simulations of Arctic mixed-phase clouds during the M-PACE campaign using single column mode of CESM2 CAM6 model. They concluded that the simulations using aerosol-aware ice nucleation schemes and including SIP processes resemble the observed single-layer mixed-phase clouds during the M-PACE. In these simulations, SIP plays a key role, and there is a competition between ice nucleation and SIP. Overall, the manuscript is well organized, and the logic is clear. However, there are several concerns that should be clarified before considering the manuscript for publication. The reviewer would recommend major revision for this manuscript in case the authors need more time for revision.

**Reply:** We thank the reviewer for the positive comments. We have revised the manuscript following your suggestions regarding the quantitative analyses and clarified the text to improve the quality of our paper.

**Major comments:**

1. Analyses: The analyses in the manuscript are full of qualitative phrases. Some examples are listed in the minor comments. Please conduct quantitative analyses.

**Reply:** We thank the reviewer for the suggestion. We have conducted quantitative analyses and improved the qualitative phrases in the revised manuscript.

2. How did the authors attain the simulated ice crystal number concentration (ICNC) for comparison with observations? Did the authors consider snow particles? Because observations should include all types of ice particles, the authors should include all ice categories for comparison. Meanwhile, in the comparison only the observed ICNC with sizes larger than 100 microns are considered, while the entire size range of simulated ICNC is used. So, the comparison is also unfair. Please use the same size range of all types of ice particles for comparison.

**Reply:** We thank the reviewer for the comments. The simulated ice crystal number
concentration (ICNC) includes both cloud ice and snow particles, for a consistent
comparison with observations. We have added a sentence in the revised manuscript as:
"Since the measurements cannot distinguish snow from cloud ice, the simulated ICNC,
IWP, and IWC all include the snow component for the comparison with observations."
Following the reviewer's suggestion, we have replotted Figure 3 in which only ice particles
larger than 100 microns are used from simulations, shown as Figure R1 below. The purpose
of Figure 3 is to examine the relative importance between primary ice nucleation and SIP
by comparing INP and ice number concentrations (not comparing simulated and observed
ICNC). The idea is, INPs represent the primary nucleated ice, and the difference between
INP and total ice number concentrations reflects the contribution of SIP. Therefore, we
used all sizes of ice crystals in Figure 3. We have added a note in the Figure 3 caption:
"The purpose of this figure is to examine the relative importance between primary ice
nucleation and SIP by comparing INP and ice crystal number concentrations. Therefore,
all ice sizes are included in the simulation results".
We have Figure 4 in the manuscript which is specifically aimed at comparing the simulated
and the observed ice number concentrations. Figure 4 already uses the simulated ice larger
than 100 microns, so we do not modify Figure 4.

[Figure]

Figure R1. Same as Figure 3 but only shows ice particles with diameters larger than 100 µm from simulations.

**Reply:** We thank the reviewer for the comment. The observed ICNC data we used in this study do not remove the shattering effect during the data quality control, since the ICNCs for M-PACE were measured before anti-shattering algorithms were developed to remove the shattered particles for the 2DC cloud probes. We contacted the data collector Dr. McFarquhar to confirm this. At his suggestion, we applied a factor of ¼ to the M-PACE observed ICNCs.

We have conducted a sensitivity test with a scaling factor of 1/2 to the observed ICNCs, as shown in supplementary Figure S3 (attached below as Figure R2). The conclusion of model and observation comparison of ICNCs is not sensitive to this correction factor. We added some discussions about this sensitivity test in the main text: "A different scaling factor of 1/2 is applied to the observed ICNCs, which increases the observed ICNCs by a factor of 2 (Figure S3). The underestimation of ICNCs by the model experiments with only ice nucleation (CNT, N12 and D15) is even worse and our conclusion regarding model and observation comparison of ICNCs is not changed."

[Figure]

Figure R2. Same as Figure 3a and Figure 4a, but applied a correction factor of 1/2 to the measured ice crystal number concentrations for Figure 3a (left) and Figure 4a (right).

**Reply:** We thank the reviewer for the great comment. Although these three schemes differ in details about temperature (and aerosol) dependences (Figure 3), CNT, N12, and D15

predict much lower INP concentrations for the M-PACE single-layer clouds than those from the B53 and M92 schemes. With these low INP concentrations, modeled clouds are overwhelmingly dominated by liquid-phase (Figures 1, 2, and 6). Therefore, it is not surprising to see the overall similar cloud states among CNT, N12, and D15. For comparison, B53 and M92 which are only dependent on temperature and not limited by aerosols predict much higher INP concentrations. With these high INP concentrations, modeled clouds with the B53 and M92 schemes are dominated by ice-phase.

We have added a note when we discuss about Figure 2 in the revised manuscript:
"Although these schemes differ in details about temperature and aerosol dependences (Figure 3), CNT, N12, and D15 predict much lower INP concentrations during M-PACE than those from the B53 and M92 schemes. With these low INP concentrations, the single-layer clouds modeled with the CNT, N12 and D15 schemes have similar cloud states (e.g., dominated by liquid-phase) (Figures 1 and 2). In contrast, B53 and M92 which are only dependent on temperature and not limited by aerosols predict much higher INP concentrations. With these high INP concentrations, modeled clouds with the B53 and M92 schemes are dominated by ice-phase."

5.      It is not clear that how the authors attained the INP number concentrations from observations and simulations especially for B53 scheme. Did the author conduct a fair comparison between them? Did the authors include all types of ice nucleation for comparison? Please provide a more detailed description.
**Reply:** We thank the reviewer for the questions. The INP number concentrations were measured by a CFDC on board an aircraft (Prenni et al., 2007) during the M-PACE single-layer mixed-phase cloud period. For the B53 scheme in the model, we use Equation 4 to calculate the immersion freezing rate, and diagnose INP number concentrations by multiplying the immersion freezing rate by the model timestep. The contact ice nucleation is based on Young (1974), and deposition ice nucleation on Meyers et al. (1992) in the model simulation. We include all these types of ice nucleation for the comparison with observations. However, for the single-layer mixed-phase clouds, the immersion freezing is dominated, and the contributions from deposition and contact ice nucleation to total ice production are much smaller (see Figure R3 below).

We have provided a more detailed description in section 3: "The N12, D15, B53, and M92 experiments are the same as the CNT experiment except using the respective ice nucleation scheme to replace the CNT scheme for the immersion freezing (section 2.2). The deposition and contact ice nucleation are still based on the CNT scheme in the N12 and D15

experiments, or based on Meyers et al. (1992) and Young (1974), respectively in the B53

and M92 experiments."

In section 4 (Results) we added: "The contributions from deposition and contact ice nucleation to total ice production are much smaller compared to the immersion freezing for the single-layer mixed-phase clouds during M-PACE."

[Figure]

Figure R3. Vertical profiles of (a) ice production rate (unit: kg kg$^{-1}$ s$^{-1}$) from immersion freezing of cloud water, (b) ice production rate (unit: kg kg$^{-1}$ s$^{-1}$) from contact freezing of cloud water, and (c) ice production rate (unit: kg kg$^{-1}$ s$^{-1}$) from deposition nucleation calculated in the B53 and B53_SIP experiments.

6.     "Section 4.3 Interactions between PIP and SIP": SIP suppressed the PIP. Did the authors consider whether some setups in the microphysics scheme lacking physical meaning result in or enhance this suppression? For example, suppression is due to decreasing difference between total ice nucleation number from parameterization and increasing ice particle number. Please provide a discussion.

**Reply:** We thank the reviewer for the good question. We understand that the reviewer is talking about the ice nucleation tendency calculated as the difference between total ice nucleation number from parameterization and ice particle number at current model time step. This tendency is reduced when the current time step ice particle number is increased due to SIP. However, the ice production rates (for ice mass) from ice nucleation shown in

Figure 10 are directly calculated by the CNT ice nucleation parameterization, which are the number of ice crystals nucleated from the parameterization times the initial mass of an ice particle (2.093×10$^{-15}$ kg). As we explain in the text, the suppression of PIP by SIP is due to lower number concentrations of INPs and cloud droplets after considering SIP.

7. Some "rate"s in the manuscript are confusing. If the reviewer understood correctly, the production rates in the manuscript are mainly for ice mass based on Figures 8-10. The question is how IIC increases ice mass? The "ice" in the manuscript all means "cloud ice" and does not include "snow"? If yes, following comment #2, different categories of ice are defined artificially in microphysics schemes, and it might not be true in observations. The authors should clarify it. The reviewer would recommend conducting analyses including simulated snow particles.

**Reply:** We thank the reviewer for the suggestion. Yes, the production rates in Figures 8-10 are for ice mass, which are calculated from ice production rates for ice number from the parameterizations multiplied by the initial mass of an ice particle ($2.093 \times 10^{-15}$ kg). We added a note in the text: "The ice mass production rates are calculated by multiplying ice number production rates from parameterizations by the initial mass of an ice particle ($2.093 \times 10^{-15}$ kg)."

In all analyses for the comparison of modeled ICNCs, IWP, and IWC with observations, modeled cloud ice and snow are added together. We agree with the reviewer that cloud ice and snow are separated artificially in the microphysics scheme in the model. IIC represents the process that snow particles collide with each other and produce smaller cloud ice particles due to the snow fragmentation. In Figure 10c, the IIC process rate indicates an increase in cloud ice mass from the fragmentation of colliding snow particles. Ice mass is converted from snow to cloud ice in the IIC process, although the total ice mass is not changed.

**Minor comments:**

1. Lines 107-108: Please describe how the graupel mass and number are diagnosed in the scheme briefly.

**Reply:** We thank the reviewer for the comment. We diagnose the graupel mass based on cloud water, cloud ice, and snow mass mixing ratio. We have added the diagnostic method in the revised manuscript as:

"The graupel mass mixing ratio ($q_g$) is diagnosed as the precipitation ice mass (currently snow, $q_s$) multiplied by the rimed mass fraction $Ri$ (Zhao et al., 2017),

$$q_g = q_s \times Ri \tag{6}$$

The rimed mass fraction $Ri$ is calculated as:

$$Ri = \frac{m_{rimed}}{m_{rimed} + m_{unrimed}} \approx \frac{1}{1 + \frac{6 \times 10^{-5}}{q_c(q_i + q_s)^{0.17}}} \tag{7}$$

$q_c$, $q_i$, and $q_s$ in (7) are modeled cloud water, cloud ice, and snow mixing ratios (kg kg$^{-1}$),
respectively. The graupel number is assumed to have the same ratio to snow number as the
ratio of graupel mass to snow mass."

2.    Lines 209-225: Please quantify the analyses, e.g., percentage of enhancement,
reduction, "largest", "smallest", "modest", "closest", "significantly
decreases/increases", …
**Reply:** We thank the reviewer for the suggestion. We have modified the sentences as:
"In the SIP experiments with the CNT, N12, and D15 ice nucleation schemes, simulated IWP
is increased from 5 to 10 g m$^{-2}$ and LWP is decreased from 156 to 97 g m$^{-2}$ averaged over
the M-PACE period after considering the SIP. In the SIP experiments with the B53 and M92
schemes, however, SIP has a minimal impact on the LWP/IWP. Second, the B53, B53_SIP,
M92, and M92_SIP produce the largest IWP (~12 g m$^{-2}$ averaged over the M-PACE period),
followed by CNT_SIP, N12_SIP, and D15_SIP (~10 g m$^{-2}$ averaged over the M-PACE
period). CNT, N12, and D15 experiments produce the smallest IWP (~5 g m$^{-2}$ averaged over
the M-PACE period). These characteristics are also evident in the vertical profiles of LWC
and IWC in Fig. 2 and Fig. S2. It indicates that the B53 and M92 nucleation schemes are
highly efficient in forming ice; in comparison, the SIP simulations using CNT/N12/D15 ice
nucleation schemes show the lower ice production capabilities. B53, B53_SIP, M92, and
M92_SIP experiments generate the closest IWP (~12 g m$^{-2}$ averaged over the M-PACE
period) compared with the observation (~64 g m$^{-2}$). However, these four experiments also
show substantially low biases of LWP (~40 g m$^{-2}$ compared with 126 g m$^{-2}$ in the observation
averaged over the M-PACE period). As shown in Fig. 1 and Fig. S1, the mixed-phase clouds
are almost fully glaciated during the single layer stratus period. Therefore, the CNT_SIP,
N12_SIP, and D15_SIP experiments give the best simulation results in terms of LWP and
IWP during the M-PACE. Adding the SIP does not change the modeled LWP/LWC and
IWP/IWC with the B53 and M92 ice nucleation schemes. On the contrary, SIP decreases the
LWP/LWC by 38% and doubles the IWP/IWC with the CNT, N12, and D15 ice nucleation
schemes."

3.    Lines 233-234: "appears an inversely linear relationship", "this relationship is not
as clear", do they have statistical significance?
**Reply:** We thank the reviewer for the comment. The purpose of this figure is to compare
$N_{INPs}$ with ICNCs, not to derive a relationship between $N_{INPs}$ and temperature. We have
removed the word "linear" and revised the related sentence as: "With the empirical ice nucleation schemes (e.g., N12 and D15), there appears an inversely relationship between $\log_{10}(N_{INPs})$ and temperature".

4.      Lines 234-238: Please quantify the analyses, e.g., "reduces dramatically", "much higher", …

**Reply:** We thank the reviewer for the suggestion. We have revised the sentences as: "However, this relationship is not as clear with the CNT and B53 schemes, and $N_{INPs}$ reduces rapidly at temperatures warmer than -15 ºC, from ~$10^{-1}$ $L^{-1}$ at $-17$°C to <$10^{-5}$ $L^{-1}$ at $-13$°C (Fig. 3b, e). In contrast, $N_{INPs}$ with the aerosol-independent M92 scheme is less variable with temperature, and is 1-7 orders of magnitude higher than that with the aerosol-aware schemes"

5.      Lines 253-264: Why is SIP not active in B53_SIP and M92_SIP? Is there a maximum threshold of ICNCs in the microphysics scheme?

**Reply:** We thank the reviewer for the comment. We understand the reviewer's concern that the inactivity of SIP in B53_SIP/M92_SIP might be caused by a maximum threshold of ICNCs imposed in the microphysics scheme. However, this is not the case in the two model experiments. We have conducted in-depth analyses and given an explanation in Section 4.3 (Figure 10). The reason for the inactive SIP in B53_SIP/M92_SIP is because of the competition between PIP and SIP (Figure 10). Too strong primary ice nucleation in B53_SIP and M92_SIP consumes available liquid cloud water, which results in less graupel in clouds. With less graupel amount, SIP through IIC is suppressed (see detailed explanation in Section 4.3 and Figure 10).

6.      Line 269: Please quantify "slightly higher"

**Reply:** We thank the reviewer for the suggestion. We have calculated the vertically integrated ice number to be $1.649\times10^6$ and $1.646\times10^6$ $m^{-2}$ in the N12 and D15 experiments, respectively. So, ice number concentrations in N12 and D15 are very similar. We have removed: "even though the N12 experiment has a slightly higher ice enhancement ratio compared with the D15 experiment."

7.      Lines 281 and 283: Please quantify "overestimate", "predominantly"

**Reply:** We thank the reviewer for the suggestion. We have provided quantitative numbers and revised the sentences as: "The CNT, N12, and D15 experiments share the similar cloud phase distribution and all overestimate the SLF in clouds with the vertically averaged SLF of 96.25%, 96.28%, and 96.26% in CNT, N12, and D15, respectively, compared to 64.35% from the observation. On the contrary, the B53 and M92 experiments with more efficient ice nucleation show predominantly ice phase clouds with the vertically averaged SLF of 17.62% and 16.43%, respectively, which agrees with previous findings (Liu et al., 2011).”

8.      Lines 287-288: How about the TWC in these simulations?

**Reply:** We thank the reviewer for the suggestion. The TWP is reduced with decreased LWP (and SLF) and increased IWP in these simulations, as shown in Table R1 below. We have added a sentence in the revised manuscript: “The TWC is reduced with the total water path (TWP = LWP + IWP) decreased from 218.5, 219.2, and 219.1 in CNT, N12, and D15 to 132.6, 131.0, and 130.8 in CNT_SIP, N12_SIP, and D15_SIP, respectively”.

Table R1. LWP, IWP, and TWP in different experiments for the single layer mixed-phase clouds period.

|         | LWP    | IWP   | TWP    |
|---------|--------|-------|--------|
| Obs     | 190.19 | 74.66 | 264.85 |
| CNT     | 217.62 | 0.93  | 218.55 |
| N12     | 218.30 | 0.95  | 219.25 |
| D15     | 218.12 | 0.97  | 219.09 |
| CNT_SIP | 129.98 | 2.58  | 132.55 |
| N12_SIP | 128.40 | 2.61  | 131.01 |
| D15_SIP | 128.19 | 2.62  | 130.81 |

9.      Lines 294-308: It is confusing whether the authors talked about ice number or mass in Figure 7. If the authors talked about ice mass in Figure 7, how do IIC contribute to ice mass?

**Reply:** We thank the reviewer for the suggestion. We are sorry for the confusion. Figure 7 shows the relative contribution from different processes to the total ice (mass) production rate. We output PIP and SIP number process rates, and multiple them by the initial mass of an ice particle ($2.093\times10^{-15}$ kg) to calculate the ice (mass) production rates used in Figure 7. We added a note in the manuscript: “The ice mass production rates are calculated by multiplying ice number production rates from parameterizations by the initial mass of an ice particle ($2.093\times10^{-15}$ kg).”

IIC represents the process that bigger snow particles collide with each other and produce smaller ice particles due to fragmentation. In the model, IIC process rate indicates a mass transfer from snow to cloud ice. It is true, the total ice mass is not changed, but ice mass is transferred from snow to cloud ice in the model, which separates the total ice into cloud ice and snow categories.

10. Line 328: Please quantify "substantially weakened"

**Reply:** We thank the reviewer for the comment. We have revised the sentence as: "The immersion ice nucleation is weakened by a factor of 4.5 (Fig. 9a) …"

11. Lines 342-343: Based on Eq. (5), M92 seems dependent on supersaturation not temperature and cloud droplet number concentration.

**Reply:** We thank the reviewer for the comment. We are sorry for the confusing. M92 is dependent on ice supersaturation. Since the model microphysics assumes saturation vapor pressure with respect to liquid in mixed-phase clouds to calculate ice supersaturation (i.e., $(e_{sl}-e_{si})/e_{si}$, $e_{sl}$ and $e_{si}$ are the saturation vapor pressures with respect to liquid and to ice, respectively), M92 is indirectly dependent on temperature. In the model, if there are no cloud droplets, ice nucleation will not occur. Thus, M92 also depends indirectly on cloud droplet number concentration.

12. Lines 362-367: Please quantify the analysis.

**Reply:** We thank the reviewer for the suggestion. we have revised the related sentences as "A smaller graupel-related IIC rate (with the peak value of 2 kg kg$^{-1}$ s$^{-1}$) (Fig. 10f) in M92_SIP compared to CNT_SIP (with the peak value of 10 kg kg$^{-1}$ s$^{-1}$) is a result of smaller graupel mass mixing ratio in M92_SIP (with the peak value of 1.4 mg kg$^{-1}$ in M92_SIP versus 5.2 mg kg$^{-1}$ in CNT_SIP) (Fig. 10g). As the graupel mass is diagnosed from the cloud water mass, snow mass, and temperature, smaller mass mixing ratios of cloud water (with the peak value of 8 versus 125 mg kg$^{-1}$ in Fig. 10h) and snow (with the peak value of 1.4 versus 2.3 mg kg$^{-1}$ in Fig. 10i) in M92_SIP eventually lead to a smaller graupel mass mixing ratio and a smaller graupel-related IIC rate. Similar results can be found with the other ice nucleation schemes."

13. Figure 1: Please provide uncertainties of these observations.

**Reply:** We thank the reviewer for the suggestion. We have revised Figures 1, 2, S1, and S2 to include uncertainties (standard deviations) of these observations. The revised Figures 1 and 2 look:

[Figure]

Figure R4.

[Figure]

Figure R5.

14.     Figure 4: How did the authors determine the cloud top and cloud base for
observations and simulations?

**Reply:** We thank the reviewer for the suggestion. The observation data are from
McFarquhar et al. (2007), and they have already determined the cloud top and cloud base
for observation data we use in this study. More information can be found in the data
description paper (McFarquhar et al., 2007). For our model analysis, we assume that clouds
exist when LWC+IWC $>10^{-6}$ kg kg$^{-1}$. From the model top to bottom, the first model layer
with LWC+IWC $>10^{-6}$ kg kg$^{-1}$ is the cloud top, and similarly the last model layer with
LWC+IWC $>10^{-6}$ kg kg$^{-1}$ is assigned as the cloud base.

15.     Figure 5: x-axis in (h), "CTL" -> "CNT"? What are the bin sizes for x and y
variables?

**Reply:** We thank the reviewer for catching the typo in Figure 5h, which we have corrected.
We have used 25 bins for x and y variables. The bin size for temperature is 2 degree, and
the bin size for the ice number/ice enhancement is calculated by (maximum value -
minimum value)/25.

[Figure]

Figure R6.

16.     Figure 7: "total ice production rate", is the "production rate" for mass or number?

**Reply:** We thank the reviewer for the suggestion. We output PIP and SIP number process
rates, and multiple them by the initial mass of an ice particle ($2.093\times10^{-15}$ kg) to calculate
the ice mass production rates used in Figure 7. We added a note in the text: "The ice mass
production rates are calculated by multiplying ice number production rates from
parameterizations by the initial mass of an ice particle ($2.093\times10^{-15}$ kg)."

17.     Figure 9: "(h) accretion rate of cloud water by snow", how about the accretion of rainwater by snow?

**Reply:** We thank the reviewer for the comment. Figure 9h only shows the accretion of cloud water by snow, and does not include the accretion of rain by snow, since the purpose of this figure is to illustrate that a stronger "accretion rate of cloud water by snow" (8 vs. 2 kg kg$^{-1}$ s$^{-1}$) results in a lower cloud water amount (13 mg kg$^{-1}$) in the CNT_SIP experiment compared with that (23 mg kg$^{-1}$) in the CNT experiment.

18.     Figure 10: (c), (e), (F), IIC influences the mass mixing ratio of ice particles?

**Reply:** We thank the reviewer for the suggestion. This IIC process transfers ice mass from snow to cloud ice, although the total ice mass does not change.

**Response to Reviewer 3**

We thank the anonymous reviewer for his/her careful reading and constructive review of our paper. Our detailed responses to the comments follow. Reviewer's comments are in blue color, our responses are in black color, and our corresponding revisions in the manuscript are in red color.

Review for "Relative importance and interactions of primary and secondary ice production in the Arctic mixed-phase clouds" by Zhao & Liu

This manuscript compares the impacts of primary ice production (PIP) and secondary ice production (SIP) as well as their interactions on the simulation of multiple Arctic mixed-phase cloud microphysical and macrophysical properties observed during the M-PACE field campaign. The authors design a set of 10 simulations, 5 of which differ only in their treatment of ice nucleation schemes and the other 5 which utilize the same 5 aforementioned ice nucleation schemes but with representations of SIP via the ice-ice collisional breakup (IC) and rain droplet fragmentation (FR) mechanisms in addition to the Hallett-Mossop process which is represented in all 10 simulations. The authors find that 3 of the ice nucleation schemes that are aerosol-aware (CNT, N12 and D15) exhibit similar behaviour to each other in terms of their simulated ice crystals number concentration vertical profiles, supercooled liquid fraction (SLF), IWP, LWP and relative contributions from primary and SIP rates to the total ice production rate. They also find that these variables are also similar to each other for the other two ice nucleation schemes (B53 and M92). One of the main is that PIP and SIP actively influence each other. The authors also conclude that the aerosol-aware ice nucleation schemes with the IC and FR mechanisms represented best represent the single-layer mixed-phase clouds observed during M-PACE.

This is an interesting and valuable study at the forefront of effort to improve cold cloud microphysics in global climate models and their impact on cloud properties. There are however, a number of ways that the manuscript can be improved, particularly pertaining to the writing including the description of the model used and the experimental design, description of the observations and grammar. Overall, I recommend major revisions that are provided below.

**Reply:** We thank the reviewer for the positive comments. We have revised the manuscript following your comments regarding the writing including the description of the model used, the experimental design, and the observations to improve the quality of our paper.

**Major revisions:**

- The title is wordy and unclear. Perhaps revise to something like "primary and secondary ice production: interactions and their relative importance"?

**Reply:** We thank the reviewer for the suggestion. We changed the title as: "Primary and Secondary Ice Production: Interactions and Their Relative Importance" as the reviewer suggested.

- An interesting conclusion of this manuscript is the interaction between SIP and PIP which compete with one another. The suppression of SIP via PIP is intuitive, however, the suppression of PIP via SIP is less intuitive since one would initially expect that more ice crystals nucleated via PIP would allow more SIP. The explanation for the latter phenomenon provided in the manuscript relates to the lack of precipitation particles in B53 and M92 due to the enhanced glaciation of mixed-phase clouds. A description of the graupel scheme (which seems to be diagnostic based on line 364) the authors implemented would help the readers more clearly understand the mechanism instead of referring to Zhao et al. 2021. The mechanism of SIP and PIP suppression could also be summarized in the Abstract. Also, the discussion on lines 73-78 in the Introduction can also be elaborated on in this aspect when describing the work of Phillips et al. 2017b.

**Reply:** We thank the reviewer for the suggestion. The same as the reviewer, we initially expected that stronger PIP would allow more SIP. However, the model shows the suppression of SIP via PIP due to complex interactions between cloud microphysics processes resulting in the reduction of precipitation particles (rain and graupel).

Following the reviewer's comment, we added a description of the graupel scheme as: "The graupel mass mixing ratio ($q_g$) is diagnosed as precipitation ice mass (currently snow, $q_s$) multiplied by the rimed mass fraction $Ri$ (Zhao et al., 2017),

$$q_g = q_s \times Ri \tag{6}$$

The rimed mass fraction $Ri$ is calculated as:

$$Ri = \frac{m_{rimed}}{m_{rimed} + m_{unrimed}} \approx \frac{1}{1 + \frac{6 \times 10^{-5}}{q_c(q_i + q_s)^{0.17}}} \tag{7}$$

$q_c$, $q_i$, and $q_s$ in (7) are modeled cloud water, cloud ice, and snow mixing ratio (kg kg$^{-1}$), respectively. The graupel number is assumed to have the same ratio to snow number as the ratio of graupel mass to snow mass."

We have added the mechanism of SIP and PIP suppression in the abstract: "SIP is not only a result of ice crystals produced from ice nucleation, but also competes with the ice nucleation by reducing the number concentrations of cloud droplets and cloud-borne dust INPs. Conversely, strong ice nucleation also suppresses SIP by glaciating mixed-phase clouds and thereby reducing the amount of precipitation particles (rain and graupel)."

- An 80% contribution of SIP to total ice formation seems very large. Are these any observations to gauge how realistic this value is? Similarly, on lines 297-301, are there any observations to gauge how realistic these contributions are? Otherwise, this should be declared in the main text.

**Reply:** We thank the reviewer for the comment. We agree with the reviewer that an 80% contribution of SIP seems a large fraction. So far, we do not have observations to directly verify the contribution of SIP to total ice formation. However, observations have reported that ice crystal number concentrations are often a few orders of magnitude higher than INP number concentrations, as we discussed in the abstract. A recent study by Luke et al. (2021; PNAS) found that "the occurrence frequency of secondary ice events averaging to <10% over the 6 years ground-based radar measurements in the Arctic, but SIP has a significant impact in a local region when they do occur, with up to a 1,000-fold enhancement in ice number concentration." In our study, we compare observed INP number concentrations with observed ice number concentrations to identify the SIP process, as shown in Figure 3. We note that ice number concentrations are three orders of magnitude higher than INP number concentrations from the model simulations, and are two orders of magnitude higher from the observation, suggesting the dominant contribution of SIP to total ice formation.

We have added a declaration in Section 5 (Summary and conclusions) as: "More observation data are needed to identify the frequencies and conditions of SIP occurrence in cold clouds and its contribution to total ice formation so that the impact of SIP can be better quantified by the models."

- In addition to the graupel implementation mentioned above, the description of the ice nucleation schemes could also be described in more detail. All ice nucleation schemes appear to be implemented as immersion freezing schemes --- please confirm. How are deposition, condensation, and contact freezing represented? To be consistent with the other naming conventions used in the manuscript, I would also recommend changing "CNT" scheme to reflect the reference that was used (was it Wang et al. 2014 or Hoose et al. 2010)? The description of this scheme also does not include the equation and the units of all equations that are provided are missing.  For N12, is the dry diameter of dust particles predicted by MAM4?  For the D15 scheme, please include more information on the instruments that were used for the measurements and the location where the observations were taken from.  To be clear, are marine organic aerosols and sea salt not included as INPs in any of the parameterizations?  Please include in the description.

**Reply:** We thank the reviewer for the suggestions. The CNT scheme represents immersion, contact, and deposition nucleation separately with different equations. With many equations involved in the CNT scheme, we prefer not to include them in the paper, but refer the readers to Wang et al. (2014) and Hoose et al. (2010). The CNT scheme is formulated based on Hoose et al. (2010) and implemented in CAM5 by Wang et al. (2014) with further improvements of using a PDF of contact angle instead of a single contact angle in Hoose et al. (2010). We prefer keeping the name "CNT" in the paper since it is called in our previous studies (Shi and Liu, 2019; Zhao et al., 2021).

We have modified the sentence as: "CNT is formulated based on Hoose et al. (2010) and implemented in CAM by Wang et al. (2014) with further improvements of using a probability density functions (PDF) of contact angle instead of a single contact angle in Hoose et al. (2010)."

The N12, D15, B53, and M92 are empirical schemes for the immersion freezing of cloud droplets. Thus, for the D15 and N12 experiments, the deposition and contact ice nucleation are still represented by the CNT scheme. For the B53 and M92 experiments, the deposition ice nucleation is represented by M92 and the contact ice nucleation by the Young (1974) scheme. We understand that there is an inconsistency in the representation of deposition and contact ice nucleation in these experiments. However, for the single-layer mixed-phase clouds, immersion freezing is dominated, and the contributions from deposition and contact ice nucleation to total ice production are much smaller (Figure 9).

We have provided a more detailed description in section 3: "The N12, D15, B53, and M92 experiments are the same as the CNT experiment except using the respective ice nucleation scheme to replace the CNT scheme for the immersion freezing (section 2.2). The deposition and contact ice nucleation are still based on the CNT scheme in the N12 and D15 experiments, and based on Meyers et al. (1992) and Young (1974) in the B53 and M92 experiments."

We have included the units in all equations of the ice nucleation schemes.

Yes, for N12, the dry diameter of dust particles is predicted by MAM4.

For the D15 scheme, we have added descriptions for instruments and measurement
locations as: "D15 was developed as a combination of field campaign and laboratory data
measured by the continuous flow diffusion chamber (CFDC) and the Aerosol Interactions
and Dynamics of the Atmosphere (AIDA) cloud chamber. The field campaign data were
obtained during the 2007 Pacific Dust Experiment (PACDEX) on the NSF/NCAR G-V
aircraft over the Pacific Ocean basin (Stith et al., 2009), and the 2011 Ice in Clouds
Experiment – Tropical (ICE-T) on the NSF/NCAR C-130 aircraft flown from St. Croix,
US Virgin Islands (Heymsfield and Willis, 2014)."

No, marine organic aerosols and sea salt are not included as INPs in any of the
parameterizations. We have added at the end of section 2.2 as the reviewer suggested:
"Marine organic aerosols and sea salt are not included as INPs in any of the above ice
nucleation parameterizations".

• Lines 96-97: It would be better to clarify that this is the case for the default CAM6
model with MG2 microphysics.
**Reply:** We thank the reviewer for the suggestion. We have revised the sentence as:
"Graupel is not considered in the default CAM6 model with MG2 microphysics."

• More on the model description: line 165: What were the aerosols initialized with in
SCAM and what are the aerosol types that are represented? Line 168: what aerosol-cloud
interactions are represented?   g. Twomey, Albrecht, glaciation indirect effect, etc.?   Lines
171-172: can the cloud-borne aerosols released as interstitial aerosols be
reactivated?   Were the simulations not free-running or nudged to MPACE meteorology?

**Reply:** We thank the reviewer for the comments. The SCAM is initialized with monthly
averaged aerosol concentration profiles for a given location, which are derived from a
present-day CAM6 climatological simulation. The initialized aerosols and precursor gases
include dust, sea salt, black carbon (BC), sulfate, particulate organic matter (POM),
secondary organic aerosol (SOA), $SO_2$, dimethyl sulfide (DMS), and a lumped condensable
organic gas species (SOAG).
In the model, Twomey, Albrecht, and INP glaciation indirect effects are represented in the
model (Liu et al., 2012; Ghan et al., 2012). Yes, the cloud-bore aerosols released as interstitial aerosols can be reactivated when clouds form. The simulations are nudged to M-PACE meteorology.

We have made the corresponding changes in the revised manuscript: "In SCAM, aerosols are initialized with monthly averaged profiles for different aerosol types (sulfate, BC, particulate organic matter, secondary organic aerosol, dust, sea salt) at a given location, which are derived from a present-day CAM6 climatological simulation."
"The cloud-borne aerosols will be released to the interstitial aerosols once cloud droplets evaporate, which can be re-activated when cloud droplets are nucleated."

- Line 194: please cite the original source of the observations. The ground-based observations are not directly comparable with the model and should be stated.

**Reply:** We thank the reviewer for the comment. We added the original sources of the observations: "Dong and Mace, 2003; Shupe et al., 2005; Deng and Mace, 2006; Turner et al., 2007; Wang, 2007; Khanal and Wang, 2015"; "We note that these data may not be directly comparable with the model outputs" in the revised manuscript.

- Line 200: Dividing by a factor of 4 seems very approximate to account for shattering effects. I would suggest using a dataset that has been revised according to the interarrival times for more accurate comparisons (Korolev et al. 2015)

**Reply:** We thank the reviewer for the constructive comment. We agree with the reviewer that "Dividing by a factor of 4 seems very approximate to account for shattering effects". We adopted this method since the M-PACE data were collected before the advent of shatter mitigating tips and before algorithms for removing the shattered particles had been developed. Thus, there were no corrections for the shattering effects on these data. We discussed this issue with Greg McFarquhar who collected the M-PACE data. He suggested that we can get some estimates of the magnitude of the shattering effect on ice number concentrations from other campaigns, such as ISDAC, IDEAS-2011, and HOLODEC, which also used the 2DC cloud probe, but adopted anti-shattering tips and algorithms for removing the shattered particles.

Previous studies indicated a reduced ice number concentrations by 1-4.5 times and up to a factor of 10 depending on particle size for IDEAS-2011 and ISDAC after using the anti-shattering tips (Jackson and McFarquhar, 2014; Jackson et al., 2014). Figure 10 in Jackson et al. (2014) below indicates that the shattering effect increases the ice number by 1-4.5 times, and the effect is stronger for smaller ice than larger ice.

To address the reviewer's concern, we did a sensitivity test with a scaling factor of 1/2 to the observed ICNC, as shown in supplementary Figure S3. We added some discussion about this sensitivity test in the main text:

"A different scaling factor of 1/2 is applied to the observed ICNCs, which increases the observed ICNCs by a factor of 2 (Figure S3). The underestimation of ICNCs by the model experiments with only ice nucleation (CNT, N12 and D15) is even worse and our conclusion regarding model and observation comparison of ICNCs is not changed."

[Figure]

FIG. 10. (a) Mean $N(D)$ from the 2DC with standard and modified tips for 25 Oct 2011 and 1 Nov 2011. (b) Mean $N(D)$ from the 2DC with standard and modified tips for 30 Apr 2008. Error bars indicate standard deviations produced by the bootstrap technique. (c) SDs for IDEAS-2011 flight RF03, 1921:30–1926:41 UTC, comparing the HOLODEC and standard and modified 2DC instruments.

From Jackson et al. (2014), Figure 10.

- Why don't B53, B53_SIP, D15 and D15_SIP not appear in Figs. 1 and 2? Please include.   Please also include the observations in Fig. 2.

**Reply:** We thank the reviewer for the suggestion. We have put B53, B53_SIP, D15, and D15_SIP results in Figs. S1 and S2 in the manuscript. Otherwise, Figs. 1 and 2 will be too busy, as current Figs. 1 and 2 have already had 6 lines and five makers. We have added the observations in Fig. 2 as:

[Figure]

- Fig 5: I find the "enhancement ratio" confusing because the relative enhancement in Figures b-j are compared relative to Figure a, but they all use the same colour bar. Wouldn't it make more sense to use a separate colour scheme for b-j?

**Reply:** We thank the reviewer for the comment. We however, find that it is hard to include two color schemes in Fig. 5. Since we are plotting the bivariate joint probability density functions (PDF) for all the panels, we think that it would be cleaner to use the same color scheme and thus keep Figure 5 unchanged.

- Please include error bars in the observations and all simulations.

**Reply:** We thank the reviewer for the suggestion. We have added error bars in the observations and all simulations in Figures 1 and 2, which are shown below.

[Figure]

[Figure]

**Minor revisions:**

•      Line 12: "of" needed after "importance"

**Reply:** Thanks. We have changed the sentence to: "the interactions between primary and

SIP processes and their relative importance…"

•      Line 32-34: another source of ice particles in mixed-phase cloud could be from ice crystals that fell from overlying cirrus clouds.

**Reply:** We thank the reviewer for the suggestion. We have added a sentence to discuss the seeding effect as: "Ice crystals that fall from overlying cirrus clouds can provide another source of ice in mixed-phase clouds."

- Lines 42-43: Ice crystal fall speed is a cloud microphysical process that is also quite important for mixed-phase cloud properties such as SLF according to the CAM5 model shown by Tan & Storelvmo 2016.

**Reply:** We thank the reviewer for the suggestion. We have modified the sentence as: "In addition, other microphysical processes such as rain formation, ice growth, and ice sedimentation are important for mixed-phase cloud properties (Mülmenstädt et al., 2021; Tan and Storelvmo, 2016)".

- Line 70: "Albeit these studies, how…" is grammatically incorrect.

**Reply:** We thank the reviewer for the suggestion. We have revised the sentence as: "Despite the above progress, many questions remain unexplored for the Arctic mixed-phase stratus clouds, e.g., whether PIP always promotes the SIP and how SIP influences the PIP."

- Line 188: "rather than" I think should be "in addition to" since Hallett-Mossop is included in all simulations?

**Reply:** Corrected. Thanks.

- Line 248: suggest replacing "in accompany with" with "accompanied by" and again on line 409.

**Reply:** Corrected. Thanks.

- Line 370: add "rate" after "nucleation"

**Reply:** Added. Thanks.

- Lines 423-426: Not necessary to discuss here since there is no associated figure and discussion and not central to the manuscript?

**Reply:** We thank the reviewer for the suggestion. These sentences are removed in the revised manuscript.